# A transdisciplinary, comparative analysis reveals key risks from Arctic permafrost thaw
Susanna Gartler [1,2,27] ✉, Johanna Scheer [3,4,27] ✉, Alexandra Meyer [1,2,27] ✉, Khaled Abass [5,6], Annett Bartsch [2,7], Natalia Doloisio[8], Jade Falardeau [9], Gustaf Hugelius [10], Anna Irrgang [11], Jón Haukur Ingimundarson[12], Leneisja Jungsberg [13], Hugues Lantuit [11,14], Joan Nymand Larsen [12], Rachele Lodi [15,16], Victoria Sophie Martin[2,17], Louise Mercer[18], David Nielsen [19,20], Paul Overduin [11], Olga Povoroznyuk [1,2], Arja Rautio[6], Peter Schweitzer [1,2], Niek Jesse Speetjens[21,22], Soňa Tomaškovičová[4], Ulla Timlin[23], Jean-Paul Vanderlinden[24,25], Jorien Vonk [21], Levi Westerveld[26] & Thomas Ingeman-Nielsen[4]

Permafrost thaw poses diverse risks to Arctic environments and livelihoods. Understanding the effects of permafrost thaw is vital for informed policymaking and adaptation efforts. Here, we present the consolidated findings of a risk analysis spanning four study regions: Longyearbyen (Svalbard, Norway), the Avannaata municipality (Greenland), the Beaufort Sea region and the Mackenzie River Delta (Canada) and the Bulunskiy District of the Sakha Republic (Russia). Local stakeholders' and scientists' perceptions shaped our understanding of the risks as dynamic, socionatural phenomena involving physical processes, key hazards, and societal consequences. Through an inter- and transdisciplinary risk analysis based on multidirectional knowledge exchanges and thematic network analysis, we identified five key hazards of permafrost thaw. These include infrastructure failure, disruption of mobility and supplies, decreased water quality, challenges for food security, and exposure to diseases and contaminants. The study's novelty resides in the comparative approach spanning different disciplines, environmental and societal contexts, and the transdisciplinary synthesis considering various risk perceptions.

The Arctic permafrost, home to more than three million people[1], forms the foundation of human life and is a crucial component of coupled socio-ecological systems[2]. Arctic permafrost is, however, warming and thawing[3–5], and projections indicate that most of it will degrade and disappear by 2050[1]. Driven by climatic and environmental changes, as well as human disturbances, permafrost thaw poses considerable risks with far-reaching implications for the global climate system and local Arctic communities. These risks, in conjunction with rapid socioenvironmental transformations[6–9] and competing geopolitical interests[10], necessitate urgent understanding and action. At the global scale, the release of greenhouse gases from thawing permafrost creates a feedback loop that exacerbates climate warming and perpetuates permafrost degradation[11–14]. Regionally and locally, permafrost thaw leads to physical, chemical, and biological shifts and landscape and ecosystem alterations[6,7,15], which often result in hazards. These hazards, defined as harmful phenomena with adverse impacts, significantly affect Arctic communities' livelihoods[16–19] and nearly all aspects of

human life, including the economy[20], infrastructure[21–23], culture and heritage[24–27], fisheries[28,29], food and water security[17,30,31] and health[32–36]. Such complex interrelations and sequences of events constitute risks that are perceived differently among (i) individuals (e.g., scientists, local stakeholders) on the basis of their worldviews, needs, and concerns[37], as well as (ii) Arctic communities due to their differences in historical, cultural, environmental, and socioeconomic settings[19,38]. Risk perceptions ultimately influence decision-making and the implementation of the mitigation and adaptation strategies needed for local risk management. In this context, comprehensive assessments that consider the multifaceted aspects of permafrost thaw risk and diverse perceptions are essential tools for informing policymaking.

Risk assessment is the process of systematically identifying, analyzing, and evaluating (qualitatively or quantitatively) risks. In the scientific literature, risk definitions and assessment methods differ greatly, focusing either on the physical or social dimensions of risk and rarely considering its

subjective nature through perceptions[39]. In addition, the growing body of knowledge on climate-related hazards and risk assessments in the Arctic thus far consists mostly of sectoral studies[40–45], which lack a comparative approach. The risks and impacts of permafrost thaw on coupled socio-ecological systems[46] remain understudied from an inter- and transdisciplinary perspective[47–49]. The fact that risks are not perceived or understood uniformly, neither by local stakeholders nor across scientific disciplines, underscores the importance of developing a comprehensive and transdisciplinary understanding of both the environmental and societal implications of permafrost thaw. This understanding is crucial for addressing the challenges posed by permafrost degradation while considering both the unique and shared challenges faced by Arctic communities in the context of climate change.

To bridge these gaps and answer our main research question—what are the local risks from Arctic permafrost thaw?—we present a holistic, comparative, inter-, and transdisciplinary[47,48] framework and analyze permafrost thaw risks. In our framework, we consider risks holistically as a dynamic and evolving socionatural phenomenon shaped by perceptions. Risk is specifically defined as the potential occurrence of a hazard resulting from physical processes, the severity of its consequences for humans and ecosystems, and the associated perceptions, that is, the importance assigned to the said risk by stakeholders. Permafrost thaw risks are thus characterized by the relationships among the three components described as follows: (i) physical

processes, i.e., climatic, environmental, and anthropogenic processes contributing to or resulting from permafrost thaw; (ii) hazards, i.e., dangers set at the intersection of the natural and societal realms; and (iii) societal consequences, i.e., perceived effects or outcomes resulting from a hazard and impacting various life domains such as health, recreation, the economy and ecosystems. The importance of physical processes in triggering hazards and the importance of hazards in impacting life domains is assessed by integrating scientific and nonscientific perceptions[39].

Our risk assessment framework was implemented in four Arctic regions: Longyearbyen (Svalbard, Norway), the Avannaata municipality (Greenland), the Beaufort Sea region and the Mackenzie River Delta (Canada), and the Bulunskiy District of the Sakha Republic (Yakutiya, Russia) (Fig. 1). All the study areas are characterized by particular geopolitical, cultural, and socioeconomic contexts[19] as well as permafrost conditions[50–53] and are thus confronted with distinct permafrost thaw-related risks. We investigated the coupled socionatural dynamics of risks and their implications in each study area with the objective of informing local communities about prominent key hazards and consequences in their respective regions. We further identified and descriptively compared similarities and disparities across study areas and consolidated our findings to provide a composite overview of the permafrost thaw risks from all case studies. Through our comparative approach and composite overview, we ultimately aimed to generalize and increase the transferability of our

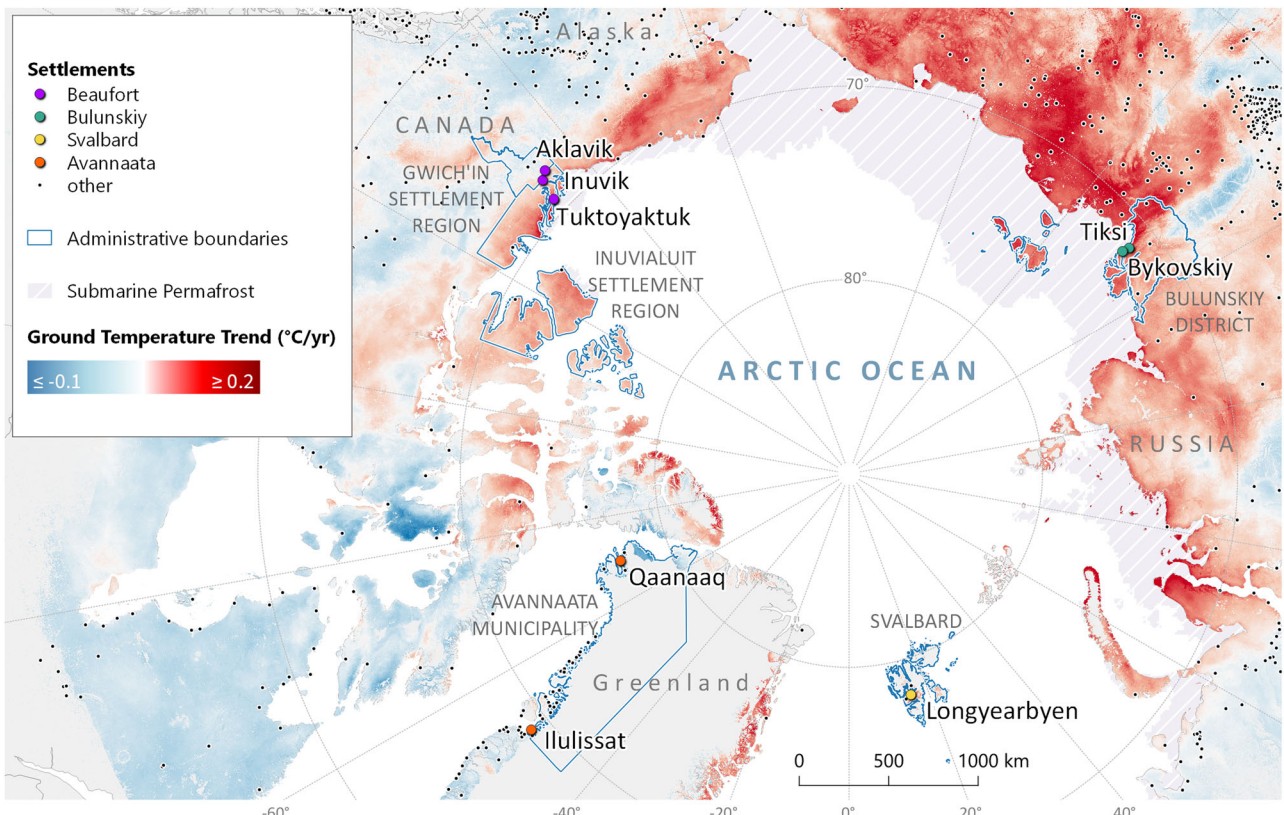

**Fig. 1 | Map of the study areas and trends in the ground temperature over the period 2000–2019.** Credits: Map by Sebastian Laboor. Arctic settlements are from the dataset Total Arctic population on settlemental level in 2017 (500+ inhab) by Nordregio[122], which is used and licensed under CC-BY-SA-4.0, and available at https://doi.org/10.1594/PANGAEA.895745. The spellings of some settlement names were edited. The submarine permafrost extent is from the Submarine Permafrost Map (SuPerMAP), which was modeled with CryoGrid 2, Circum-Arctic by Overduin et al.[123], and is used and licensed under CC-BY-4.0 and available at https://doi.org/10.1594/PANGAEA.910540. Trends in permafrost temperature were retrieved via the Center for Environmental Data Analysis from the ESA Permafrost Climate Change Initiative (Permafrost_cci): Permafrost Ground Temperature for the Northern Hemisphere, v3.0, 25 June 2021, by Obu, Westermann et al.[121], which is used and licensed under https://artefacts.ceda.ac.uk/licenses/specific_licences/esacci_permafrost_terms_and_conditions.pdf and available at https://doi.org/10.5285/b25d4a6174de4ac78000d034f500a268. The country borders are from the dataset TM_WORLD_BORDERS by http://thematicmapping.org, which is used and licensed under CC-BY-SA-3.0. The administrative borders are from OpenStreetMap[124], and are available and licensed under the Open Data Commons Open Database License (ODbL) (openstreetmap.org/copyright) by the OpenStreetMap Foundation (OSMF).

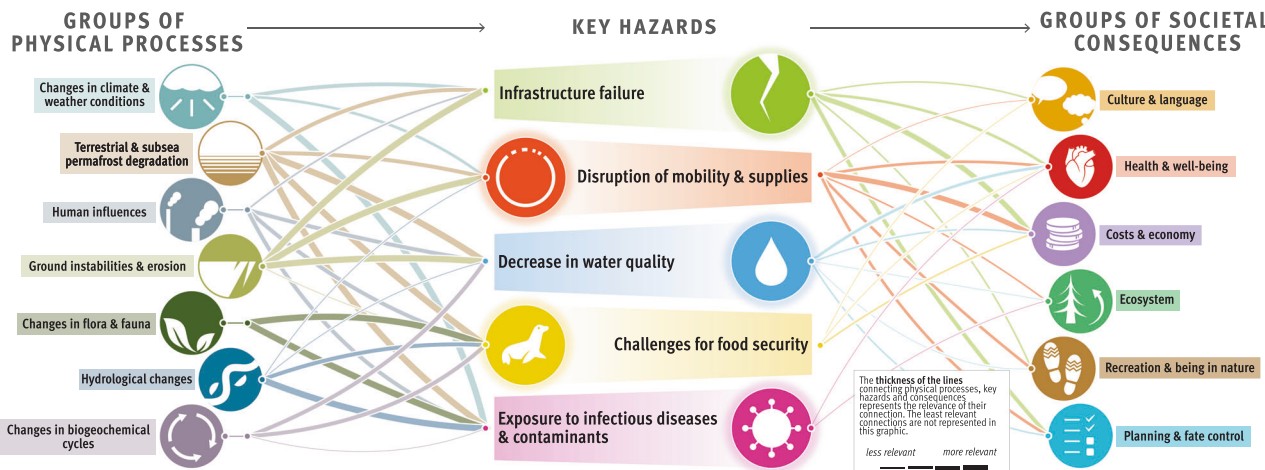

**Fig. 2 | Composite overview synthesizing the risks from permafrost thaw identified across the four study areas.** Credits: adapted from The Big Picture—It's All Connected by Westerveld et al.[125] by Johanna Scheer with the support of Levi Westerveld. The symbols to the left of the graphic represent different groups of climatic, environmental, and anthropogenic drivers of and processes resulting from permafrost thaw, which contribute to triggering harmful phenomena or events, referred to as key hazards. These key hazards, illustrated by the symbols at the center of the figure, in turn have adverse consequences for different life domains, represented by the symbols to the right of the figure. The thickness of the lines represents the rankings obtained from scientific and nonscientific perceptions integrated into our analysis (i.e., the connections between the groups of physical processes and key hazards were derived from the consortium rankings, whereas the connections between the key hazards and the groups of societal consequences were calculated by averaging the consortium and local expert rankings). The thicker the lines are, the more prominent or likely these physical processes are in triggering the hazards, and the more severe the impacts from the key hazards are for the life domains. Detailed examples of physical processes, hazards, and consequences and working definitions of these concepts are provided for each of the represented groups in Table 2.

findings to other regions, thereby supporting the development of overarching strategies for risk adaptation and mitigation.

Throughout our risk analysis, knowledge was combined and synthesized through inter- and transdisciplinary collaboration and multi-directional knowledge exchanges[54,55] involving a large variety of stakeholders, including local stakeholders[38] (local land users, Indigenous knowledge and rightsholders, administrative authorities, civic leaders, technical staff, and other experts[39]) and scientists from multiple disciplines. The risk analysis drew on primary data collection undertaken by the consortium scientists through a series of intensive fieldwork campaigns. We then gathered all available information on risks from the study areas through workshops and multiple exchanges held between 2019 and 2023, which included the participation and inputs from local Indigenous representatives in addition to over one hundred scientists from all fields within the project consortium. This information was then categorized via thematic network analysis[56], and an iterative process was adopted to rank the identified permafrost thaw risks with the consortium scientists, study region inhabitants, and (other) local experts[39]. Finally, a series of workshops and consultation meetings facilitated the risk evaluation by local experts, where the results were shared and verified together with close to one hundred stakeholders in the study areas[57]. By synthesizing information from multiple stakeholders, scientific disciplines, and four case studies from across the Arctic, we contribute to a more comprehensive understanding of the complex and interconnected factors of risk across permafrost regions and provide a knowledge basis for informed adaptation strategies and policymaking.

## Results
### Local risks of Arctic permafrost thaw: a composite overview
The composite overview in Fig. 2 presents the main findings of the risk assessment for the Arctic permafrost region, which is based on synthesized information from Longyearbyen (Svalbard, Norway), the Avannaata municipality (Greenland), the Beaufort Sea region and the Mackenzie River Delta (Canada), and the Bulunskiy District (Russia) (for more information on the study areas, see section "Description of the study areas" in Methods). The figure highlights the complex interconnectedness of physical processes, hazards, and societal consequences for life domains related to permafrost thaw. The thematic network analysis revealed five key hazards, namely, (1)

infrastructure failure, (2) disruption of mobility and supplies, (3) decrease in water quality, (4) challenges for food security, and (5) increased risk of exposure to infectious diseases and contaminants. The overview we present here focuses on these five key hazards resulting from permafrost degradation, which have far-reaching consequences for ecosystems, sociocultural dynamics, economies, governance, and the health and well-being of people in Arctic communities. The data presented in the graphic align with the temporal and spatial scales relevant to the affected communities.

Figure 2 illustrates that ground instabilities and erosion are the prevailing processes leading to infrastructure failures, disruptions of mobility and supplies, decreases in water quality, and, to a lesser extent, challenges for food security. Infrastructure failure and disruptions of mobility and supplies have the greatest impacts on costs and the economy and are generally perceived as important permafrost-related challenges in all study areas (Fig. 3). Food security is affected mostly by changes in flora and fauna, which, together with changes in climate and weather conditions and hydrological changes, also contribute to exposure to infectious diseases and contaminants. Compared with other major risks, assessing the impact of exposure to infectious diseases and contaminants has proven more challenging because of a local shortfall in expertise and information. Depending on factors such as livelihood strategies, sociocultural background, governance structures, existing infrastructure facilities, and permafrost conditions, the perception of permafrost thaw risks involves important place- and context-specific complexities[19].

The cumulative importance of permafrost hazards in causing societal consequences, as perceived and ranked by scientists and local rightsholders and stakeholders, is compared per region in Fig. 3. The figure demonstrates that there is substantial variation not only among individuals and within communities but also between regions. For example, challenges for food security (including impacts on subsistence activities) were seen as considerable permafrost thaw-related hazards with many consequences for life in the Beaufort Sea region and the Mackenzie River Delta. However, such concerns were not as prominent in Svalbard or Greenland, where infrastructure failure was perceived as the most critical hazard. In the following sections, which are ordered according to the key hazards, we further elaborate on the similarities and differences in permafrost thaw risks and their consequences across localities.

**Fig. 3 | Cumulative impact score per key hazard and study area.** The graphic depicts the importance of permafrost hazards in causing societal consequences, as perceived and ranked by the consortium and local rightsholders and stakeholders (c.f. Methods) and represented per study area. The thickness of the lines was calculated per study area by summing the scores between a given hazard and the six groups of societal consequences.

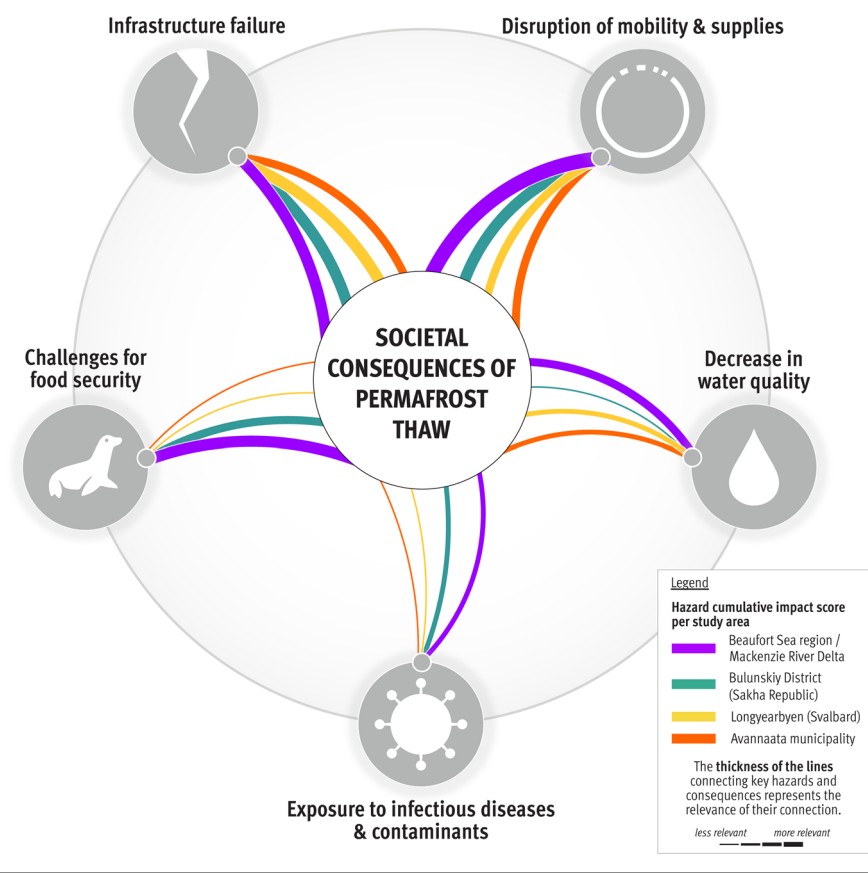

## Infrastructure failure

Arctic communities rely heavily on the services provided by their housing, communication, transportation, and energy infrastructures[21]. However, the integrity of the built environment is jeopardized by permafrost thaw and construction practices that are not adapted to the current climate. Infrastructure failures, which impact all aspects of life, were indeed a primary concern across all the study areas (Fig. 3). The significant costs incurred by repairs, the implementation of adaptation measures, and the decommissioning or relocation of infrastructures were concerning for local experts and governmental entities. For example, in Longyearbyen (Svalbard, Norway), which is located between steep mountainsides, landslides, rockfalls and ground surface deformations considerably impact buildings and infrastructure. Much of the former mining town's built environment was constructed for temporary use, resulting in a need for costly adaptation measures. In Ilulissat (Avannaata municipality, Greenland), where permafrost is relatively ice-rich and close to its thawing point, ground surface deformations and associated infrastructure damages occur seasonally. The demand for construction and maintenance operations has increased in recent years, thereby exerting pressure on the local private sector and already limited municipal budgets.

Damages to the built environment and associated adverse effects have also impacted the health and well-being of local residents in all the study areas. However, recognizing hazards has equipped individuals and communities with the ability to confront and manage them proactively, fostering a sense of control and preparedness that underpins overall well-being[35,36]. Hazardous slope-related processes, including rockfalls and landslides, notably represented a major safety concern in Longyearbyen. Similarly, landslides and tsunamis have become a growing source of concern in the Avannaata municipality, particularly following the 2017 Nuugaatsiaq disaster, which resulted in the destruction of eleven houses and claimed four lives[58]. In the Beaufort Sea region and the Mackenzie River Delta (Canada), Bulunskiy District (Russia), and Longyearbyen, erosion (fluvial or coastal) was a major concern with respect to infrastructure failure, impacts on

recreational activities, and the need for protective measures. In the Beaufort Sea/Delta region, accelerating erosion rates are leading to the complete destruction of infrastructure (Fig. 4) and contributing to the need for ongoing and planned relocation of homes, particularly in Tuktoyaktuk[59]. Coastal erosion is also threatening the cultural identity and heritage of communities in several regions, resulting in damage to remnants from the coal mining industry in Longyearbyen[60], cultural heritage sites in the Beaufort Sea/Delta region[24,61], and cemeteries in Bykovskiy (Bulunskiy District, Russia).

Given the wide range of consequences resulting from infrastructure failures, urban planning was often perceived as challenging, particularly due to the uncertainties associated with permafrost thaw. In Longyearbyen, confined development areas due to natural hazard risk zones, cultural heritage, and a state incentive to halt urban expansion represent considerable constraints for infrastructural development. In addition, turnover of staff, resulting in knowledge loss and adaptation challenges, was a shared concern across the four study areas. The multitude of tasks, difficulties in recruiting and retaining an experienced workforce, resource allocation, and prioritizing were perceived as challenging for proactive planning. In Ilulissat, local stakeholders raised concerns about sustaining the town's expansion strategies given financial and logistical constraints. Adaptation planning was generally perceived as necessary at all the study sites, but the rapidity of on-going changes challenged people's sense of fate control. In Tiksi and Bykovskiy (Bulunskiy District, Russia), the issues and concerns related to failing or potentially endangered infrastructure were not publicly discussed, and the funds to mitigate the progressing coastal erosion were lacking.

## Disruption of mobility and supplies

Thawing permafrost damages transportation infrastructure, as well as food and water supply facilities. Extreme weather events and increased erosion further disrupt navigation, limiting access to resources[18,24]. Disruption of mobility and supplies was of great concern across the study areas. In

**Fig. 4 | Cabin destruction in the Mackenzie River Delta as a result of permafrost thaw and riverbank erosion.** Credits: Picture by Angus Alunik, 2021.

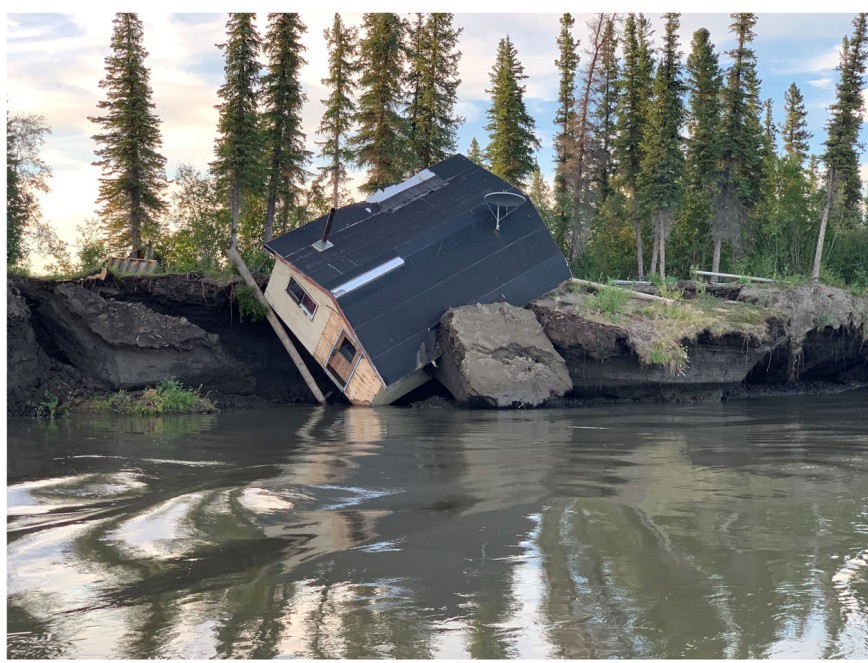

Longyearbyen, this hazard was mainly considered to be a function of infrastructure failure. Coastal erosion has damaged roads, and as the town is located in a valley with steep hillsides, permafrost thaw causes landslides and rockfalls, which have resulted in temporary road closures and disruptions of hiking paths. Furthermore, roads and the airport runway are at risk from ground surface deformations induced by permafrost thawing and ground ice melting. The likelihood of supplies being disrupted was nonetheless considered low in this study area. In the Avannaata municipality, the deterioration of roads (Fig. 5) was also a concern, primarily due to the need for frequent repairs, associated economic consequences (Fig. 6), and a lack of alternative routes to critical facilities in the case of disruption (e.g., a unique road to the airport in Ilulissat).

In comparison, disruptions of mobility and supplies occurring within the Beaufort Sea/Delta Region were perceived as having important impacts on all life domains, such as being in nature, ecosystem health, and subsistence practices, which in turn affect culture, language, and identity. Permafrost thaw results in changes in trails and river channels, complicating access to camps and creating safety challenges. Another main concern in this region was the increased costs due to road damage. In northern Sakha (Yakutiya), the degradation and destruction of winter roads, tundra trails and water pipelines and their impacts on transportation were perceived as major risks. Related delays and irregular provisioning of groceries, traditional food, and other goods challenge local food security. In addition, the formation of thermokarst wetlands or depressions (irregular topography resulting from the thawing of ice-rich permafrost and subsequent ground surface deformations[62]) in the Sakha tundra increased the number of accidents, changed the mobility patterns of local residents, and altered the migrations of reindeer and hunting game. Changing seawater turbidity and salinity drove fish resources and harvesters farther away from the coast. These new mobility patterns challenge subsistence and commercial practices, making them less safe and less productive.

## Challenges to food security

Biodiversity loss, habitat destruction, and declining animal populations pose challenges for food security in the Arctic[17]. Permafrost thaw leads to reduced travel safety, compromised trails, and the changing availability of wildlife[63]. Landscape transformations can further result in the release of contaminants, impacting subsistence and compromising food quality and supply[64]. Permafrost thaw thus constitutes a considerable risk for food security in Arctic

regions, depending, however, on the particular place-specific context. In Longyearbyen, where people rely on food imported from the mainland via air and maritime transportation, challenges for food security were not considered relevant. The study participants in Greenland considered such challenges to be unrelated to permafrost thaw. In contrast, food security was one of the main concerns related to permafrost thaw in the Beaufort Sea Region and the Mackenzie River Delta[19,65]. In this region, food security relies largely on subsistence harvesting, which is also a crucial element of cultural identity[66]. Challenges for food security were thus considered to heavily impact culture and language, as well as the economy (Fig. 7).

Similarly, in the northern Bulunskiy District (Sakha Republic), landscape transformations induced by permafrost thaw also impact traditional subsistence activities such as fishing, reindeer herding, and hunting, and associated spiritual beliefs. Furthermore, delays and irregular provisioning of groceries, traditional food, and other goods threaten local food security. Notably, a clear adverse impact of permafrost thaw is notably the loss of ice cellars (subterranean storage spaces carved into permafrost used to preserve food by maintaining low temperatures), increasing the risk of exposure to diseases through food consumption and handling of meat or fish, as well as the general loss of food stored in ice cellars and caches (especially in Bykovskiy). In both the Beaufort Sea region and the Mackenzie River Delta and in the Bulunskiy district, permafrost thaw generally had multiple indirect impacts on culture, identity, and overall community well-being, for example, through undermining traditional food production, storage, and distribution practices, and networks. In both study areas, subsistence activities shape cultural identity and contribute to health and financial independence. Additionally, economic hardship (e.g., higher costs of store-bought food) may arise from permafrost thaw impacts on food security. In the Greenlandic study area, some concerns were notably expressed regarding possibly reduced benefits from subsistence and (commercial) hunting and fishing.

## Decrease in water quality

Permafrost thaw may lead to shifts in hydrology and ground instabilities, leading to the mobilization of sediment, nutrients, and contaminants in Arctic aquatic ecosystems[28,67]. These processes affect ecosystem functions, subsistence, and access to clean water[64,68]. In Longyearbyen, a major concern related to permafrost thaw was access to clean drinking water. The main local freshwater source, Isdammen, is contained by a dam held together by

**Fig. 5 | A paved road in Ilulissat (Greenland) affected by ground surface deformations (also known as differential thaw settlements).** Credits: Picture by Johanna Scheer, 2021.

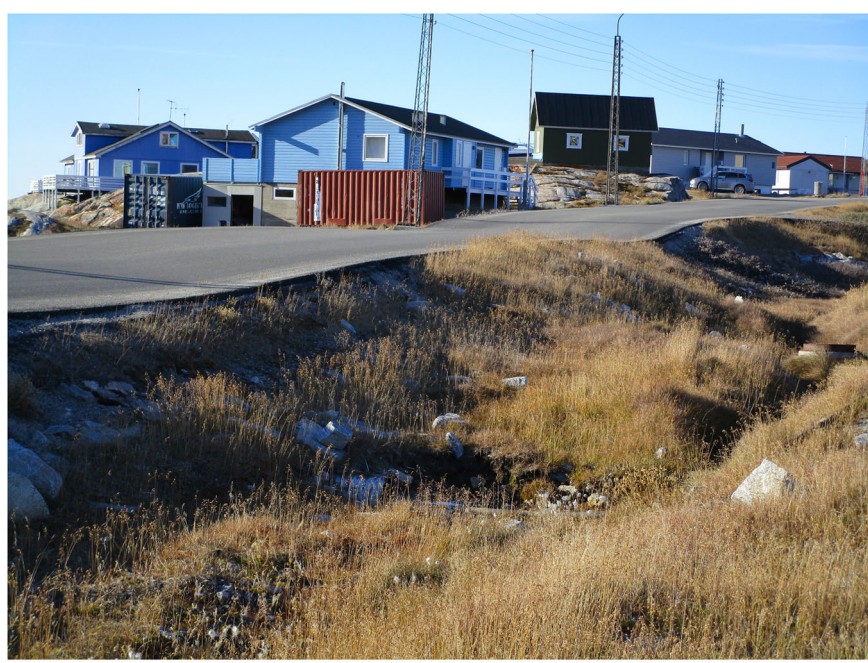

permafrost that could possibly thaw. Therefore, people were worried about the potentially severe consequences, particularly for health and well-being (Fig. 8). In contrast, in the Avannaata municipality, decreased water quality was not perceived as a very pressing concern. Although the transport of sediments and pollutants into the drinking water reservoir in Ilulissat primarily posed technical challenges attributed to the necessity of enhancing water filtration systems, adverse impacts on health were seen as a more dominant concern. Furthermore, general concerns were raised about potential water shortages in the summer. In the Beaufort Sea region and the Mackenzie River Delta, decreased water quality was also not considered a major hazard, but local experts did voice concerns about the water supply and adverse effects of permafrost thaw on aquatic ecosystems, such as changes in the quality of sea and river waters, loss of biodiversity, increased sedimentation and contamination. These effects, in turn, impact recreational and subsistence fishing and whaling. Similar perceptions were gathered in northern Sakha (Yakutiya), where decreased drinking water quality was generally not a concern, with the exception of Bykovskiy, where the eroding coastal cemetery was perceived as the main source of organic pollution both for food and water.

### Exposure to infectious diseases and contaminants

Thawing permafrost and erosion mobilize legacy contaminants and mercury (toxic metals that can harm fish, other food sources, and humans through their consumption) and may result in the spread of infectious diseases, including dormant diseases[69–72]. For example, unsecured hazardous waste can cause exposure to contaminants[73], whereas harmful algal blooms can endanger aquatic organisms[74]. In our study areas, the safety of activities such as harvesting, being in nature, engaging in outdoor recreational activities, and consuming country food was perceived to be potentially compromised. In Bykovskiy, coastal erosion severely affects cemeteries along the coastline and was therefore associated with exposure to diseases and contaminants (Fig. 9). On Svalbard, exposure to infectious diseases and contaminants due to permafrost thaw was considered low, although some experts expressed concerns regarding old landfills that might thaw and release contaminants. Exposure to infectious diseases and contaminants was ranked among the least concern in the Beaufort Sea region and the Mackenzie River Delta region. However, the study participants were worried about the uncertainty regarding (future) impacts on the ecosystem, water quality, health and well-being[75], financial security, and the economy.

Overall, we encountered at least some concern that, in all the study areas, humans and animals may be exposed directly or indirectly to infectious diseases and contaminants through food and water sources (e.g., mercury in marine ecosystems in Greenland). The release of contaminants from old oil and gas infrastructure (e.g., near Tuktoyaktuk) or from waste dumps into ecosystems was, in that respect, preoccupying. Concerns about the bioaccumulation (increase in concentrations as one moves up the food chain) of contaminants such as mercury were also raised[33]. In the Avannaata municipality and in the Beaufort Sea region and the Mackenzie River Delta, challenges related to the education, recruitment, and retention of qualified health professionals, coupled with the need to secure safe sources of food and water, were intertwined with uncertainties regarding heightened exposure to contaminants and infectious diseases.

### Conclusions

This transdisciplinary risk analysis conducted in four Arctic regions revealed the substantial impacts of permafrost thaw on the environment and livelihoods of communities. This study underscores the vital role of permafrost within Arctic ecosystems and highlights the main risks associated with its vulnerability to climate change. We assessed permafrost thaw risks by characterizing the relationships between the physical processes, key hazards, and societal consequences on life domains via thematic network analysis and how each of these relationships was perceived across the four case studies. To be considered a main risk, the connections between the physical processes and key hazards with a minimum ranking score of 1.3 (c.f. section "Step 3: Risk ranking") and the corresponding two most impacted life domains were selected. The main risks from permafrost thaw across the four regions are thus described as follows. First, infrastructure failure is a hazard caused by permafrost-related ground instabilities and erosion, human influences, and changes in climate and weather conditions, resulting in adverse consequences for costs and economy, planning and fate control, and health and well-being. Second, disruptions of mobility and supplies, often created by ground instabilities and erosion, impact costs and the economy, as well as recreation and being in nature. Third, a decrease in water quality, caused by ground instabilities, erosion, and changes in biogeochemical cycles, affects health and well-being as well as costs and the economy. Fourth, challenges for food security are created by changes in flora and fauna, hydrological changes, changes in biogeochemical cycles, and human influences, which have consequences for local costs and the

**Fig. 6 | Local risk graphic for the Avannaata municipality, Greenland.** The connections between the key hazards and groups of societal consequences were calculated by averaging the consortium and local expert rankings.

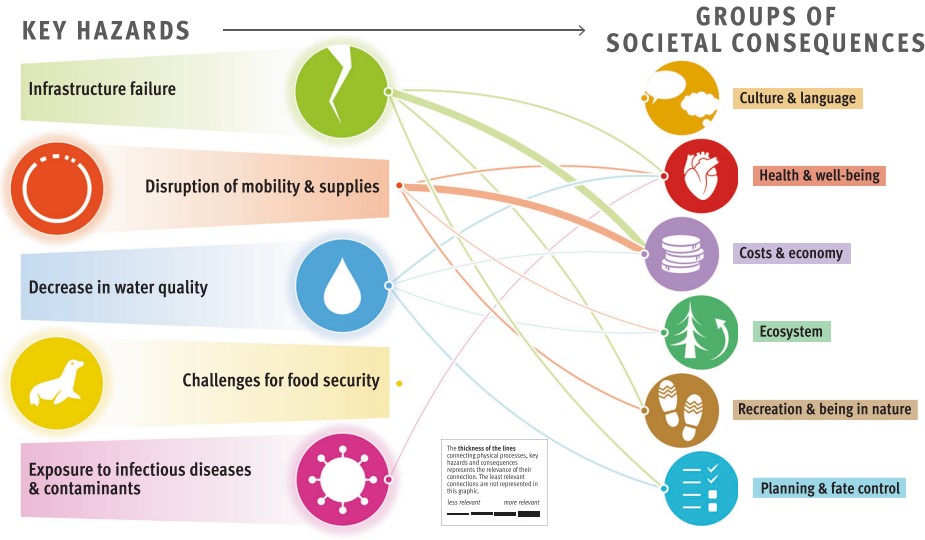

**Fig. 7 | Local risk graphic for the Beaufort Sea region and the Mackenzie River Delta, Canada.** The connections between the key hazards and groups of societal consequences were calculated by averaging the consortium and local expert rankings.

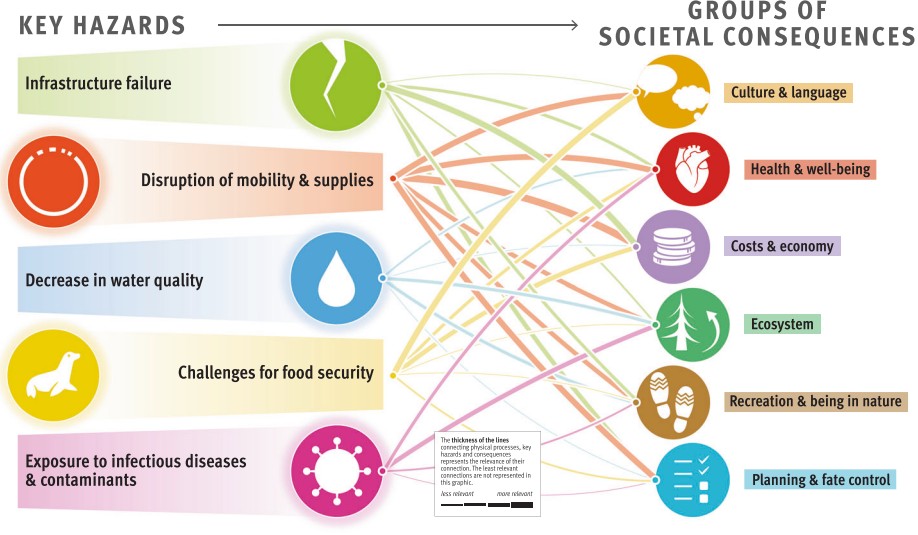

economy as well as culture and language. Finally, exposure to infectious diseases and contaminants is a hazard driven by changes in climate and weather conditions, human influences, changes in flora, fauna and hydrology, which creates concerns for human and ecosystem health and well-being.

The novelty of this study is, first, its comparative approach, which spans different environmental and societal contexts, and second, its transdisciplinary synthesis, which involves identifying risks while taking into account a large variety of stakeholders' risk perceptions. By synthesizing insights from diverse case studies and disciplines, and through continuous learning from local rightsholders and stakeholders, this study enhances our understanding of the complex and place-based factors driving risks in these landscapes. While all the study areas are characterized by continuous permafrost, they exhibit substantial place-based variations both socially (e.g., population composition, economic prosperity, and political and governance systems) and environmentally (e.g., permafrost and weather conditions, ocean proximity, topography, and vegetation). These variations contribute to the complexities in perceived risks related to permafrost thaw. Thus, while the physical processes of permafrost degradation are generally consistent across the study areas, societal consequences and concerns vary significantly due to differing environmental conditions, cultural contexts, and historical

legacies. Since our study areas represent perspectives from a range of human and natural Arctic permafrost settings, the framework and risk assessment we present here are applicable to other (continuous) permafrost regions experiencing similar hazards and impacts, thereby supporting the development of overarching adaptation and mitigation strategies. The site-specific risk analyses, in turn, inform local communities about the hazards and consequences of permafrost thaw in their respective regions (Figs. 6 to 9). In addition, integrating a more extensive and diverse database in our risk analysis allowed us to improve the accuracy and reliability of our findings. However, local concerns related to permafrost thaw are deeply entangled with other issues and processes, both those that are climate-related and those that are not, and any attempt to single out permafrost thaw-related risks necessarily involves an analytical reduction.

Arctic peoples demonstrate remarkable resilience and adaptability. Adaptation is an ongoing process, as humanity has continuously evolved to meet changing conditions. The inter- and transdisciplinary composite risk analysis presented here provides important insights into the main risks associated with permafrost thaw in Arctic coastal regions and highlights the need for proactive measures to support these adaptation and resilience efforts. By emphasizing the interconnectedness of physical processes, societal concerns, and perceptions, this research can aid policymakers,

**Fig. 8 | Local risk graphic for Longyearbyen, Svalbard.** The connections between the key hazards and groups of societal consequences were calculated by averaging the consortium and local expert rankings.

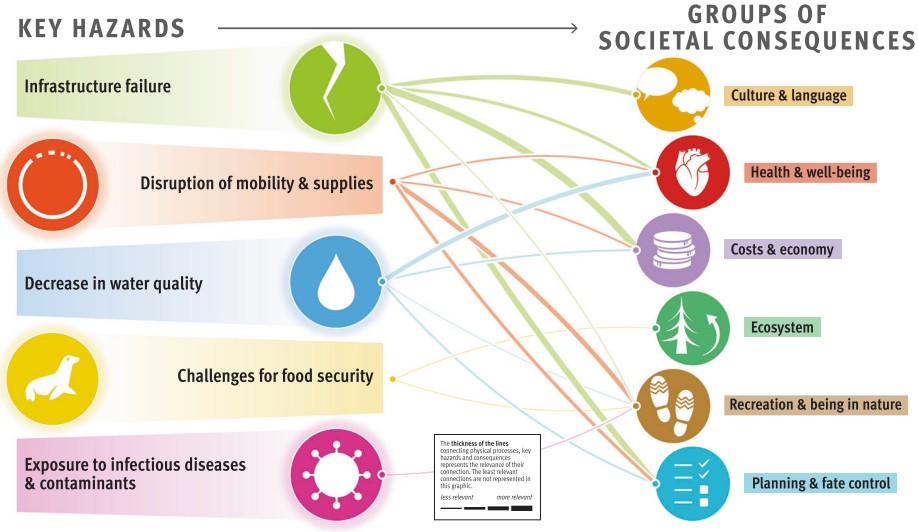

**Fig. 9 | Local risk graphic for the Bulunskiy District, Russia.** The connections between the key hazards and groups of societal consequences were calculated by averaging the consortium rankings.

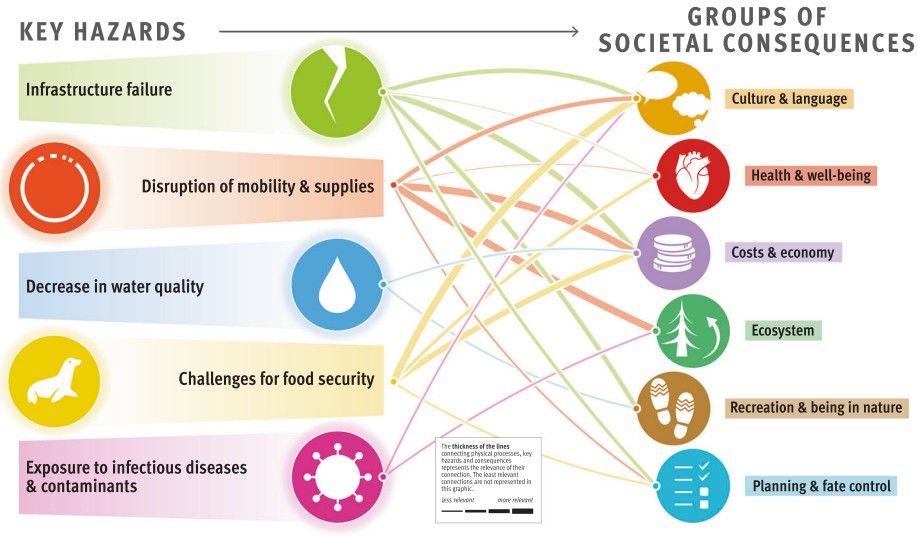

rightsholders, and stakeholders in decision-making for safer Arctic futures. This research is an example of tackling problems holistically instead of compartmentalizing interconnected issues. Accordingly, permafrost thaw risks can best be addressed by employing integrated, holistic approaches for adaptation and mitigation planning. Our research further provides a framework for future studies, which are necessary to address the inherent complexities and context-dependent nature of permafrost thaw risks and for communities to address local challenges. By knowing the key risks and hazards associated with permafrost thaw, Arctic permafrost communities everywhere can take informed actions to prevent unnecessary harm.

## Methods
### Description of the study areas
The Western Arctic Canadian case study includes four communities: Inuvik, Aklavik, Tuktoyaktuk, and Fort McPherson. Inuvik (or Inuuvik, meaning "living place"), sitting on the eastern edge of the delta, serves as an administrative and transportation hub with a population of approximately 3400[76]. The town was established in 1953 as a planned community to replace Aklavik, which was prone to flooding. Aklavik (or Aklarvik, meaning "grizzly bear place"[77]), however, persisted and is now home to approximately 700 residents[76]. Originally settled by the Inuvialuit and Gwich'in

people, it became a key location for the Hudson's Bay Company in the early 20th century. It is located in the Mackenzie Delta along the Peel Channel, approximately one hundred kilometers south of the Beaufort Sea. Most inhabitants are Inuvialuit Inuit or Gwich'in First Nation citizens, and the community has a mixed economy of wage employment and subsistence harvesting[78], similar to Fort McPherson and Tuktoyaktuk. Tuktoyaktuk (Tuktuuyaqtuuq means "resembling a caribou" in Inuvialuktun[79]) is an Inuvialuit settlement located on the shore of the Arctic Ocean. It has been a traditional hunting and fishing location for centuries and became a key point of interest during the 20th century because of its strategic location for Arctic exploration and resource extraction. Often referred to as "Tuk", the community is the terminus of the Inuvik-Tuktoyaktuk Highway and is currently home to approximately 900 residents[76]. Fort McPherson (or Tetl'it Zheh: "Town at the Head Waters") is a Gwich'in community located on the Peel River on the southern side of the Mackenzie River Delta. Established as a Hudson's Bay Company trading post in 1849, it became an essential center for fur trade. It is home to more than 700 people[76] who are predominantly Teetl'it Gwich'in First Nation citizens[80].

Inuvik and Tuktoyaktuk had mean annual air temperatures of −7.1 and −8.9 °C during the period of 1990–2020[81], respectively. Air temperatures increased at a rate of approximately 0.77 °C/decade in the region from

**Table 1 | Current trends in ground temperature and projected thaw in the study areas, retrieved via the Center for Environmental Data Analysis from ESA Permafrost Climate Change Initiative (Permafrost_cci): Permafrost Ground Temperature for the Northern Hemisphere, v3.0, 25-06-2021, by Obu, Westermann et al.[121], which is used and licensed under https://artefacts.ceda.ac.uk/licenses/specific_licences/ esacci_permafrost_terms_and_conditions.pdf, and available at https://doi.org/10.5285/b25d4a6174de4ac78000d034f50 0a268**

| Settlements | 2019 ground temperature at 2 m depth (°C) | 2000–2019 trends in ground temperature (°C/yr) | 2000–2019 linearly projected thaw at 2 m depth (yr) |
|---|---|---|---|
| Longyearbyen (airport) | −1.4 | 0.12 | 2031 |
| Qaanaaq | −7.5 | 0.06 | 2144 |
| Ilulissat | −2.5 | 0.08 | 2082 |
| Inuvik | −2.5 | 0.08 | 2050 |
| Aklavik | −1.4 | 0.06 | 2042 |
| Tuktoyaktuk | −4.3 | 0.14 | 2050 |
| Tiksi | −7.5 | 0.164 | 2065 |
| Bykovskiy | −7.3 | 0.165 | 2063 |

1981 to 2010[82]. All four communities are underlain by continuous permafrost[52]; however, the mean annual ground temperatures at a two-meter depth ranged from −1.4 °C in Aklavik to −4.3 °C in Tuktoyaktuk in 2019 (Table 1). Between 2000 and 2019, permafrost has been warming at rates reaching 0.14 °C/yr in Tuktuktoyak (Table 1) and continues to warm and deteriorate in the region[51,82,83]. Along with increasing air temperatures[82,84] and permafrost thaw, more rain, flooding, and storms[85,86], lower snow and ice levels[87], melting glaciers[88], and changing winds[89] are occurring. The region is particularly vulnerable to coastal and riverbank erosion[86,89], subsidence, and slumping[24,90,91] caused by permafrost thaw, which affects the Inuvialuit Inuit, Gwich'in the First Nation, and non-Indigenous populations. While long-term average coastal erosion rates in the region approach 0.5 m/yr[92], erosion rates of more than 10 m/yr are also reported episodically[93].

Tiksi and Bykovskiy are both located in the Bulunskiy District (ulus) of the Sakha Republic (Yakutiya). Tiksi, nestled in the Lena River delta, presently accommodates approximately 4,600 inhabitants, consisting of Indigenous groups (such as the Sakha, Evenki and Eveny) and Russian settlers. Tiksi was established in 1932 as a port city along the Northern Sea Route during the zenith of Soviet Arctic exploration, urbanization, and industrialization. Previously serving as a bustling urban settlement, transshipment node, and later, a military base, since the post-Soviet era, Tiksi has confronted population decline and infrastructural and socioeconomic quandaries. Currently, the main sectors of activity revolve around the local fishing industry, small-scale trade, and public services[94]. In contrast, Bykovskiy is a rural community with slightly more than 500 residents, situated on the Bykovskiy Peninsula, located 40 km away from Tiksi by sea. It was founded in 1940 during Soviet collectivization, leading to the sedentarization of the nomadic population. This community has evolved into the largest Indigenous (primarily Evenki) fishing community in the Bulunskiy District. Forced relocations and deportation of political prisoners in the mid-20th century added not only to ethnic diversity but also to the dark past of the community. Post-Soviet socioeconomic transformations have brought about changes in the fishing economy and intensified competition for marine resources. Nevertheless, the local fishing enterprise currently remains central to the livelihoods of the residents, offering not only employment but also basic social security and access to subsistence resources, including fish[95].

Tiksi had a mean annual air temperature of −12 °C[96] and experienced an increase in air temperature of 0.11 °C/decade during the period of 1991–2020[97]. Tiksi and Bykovskiy are both underlain by continuous permafrost with mean annual ground temperatures at a two-meter depth of −7.5 and −7.3 °C in 2019 (Table 1). While Tiksi benefits from the natural protection provided by the Bykovskiy Peninsula to the west and the coastal Kharaulakh Ridge, Bykovskiy, characterized by backshore coastal landforms such as cliffs and low-lying topographic depressions, is particularly vulnerable to erosion. Cliff morphology ranges from relatively stable vegetated slopes to nearly vertical cliffs, often exposing complexed ice and sediments or ice wedges[98]. The coastline has retreated at a mean rate of 0.59 m/yr between 1951 and 2006[98], whereas at some locations, the annual rates reached more than 10 m/yr[99]. Low-lying depressions are thermokarst basins formed by the thawing of ice-rich permafrost and subsequent surface subsidence[98].

Longyearbyen, with approximately 2600 residents, is the largest settlement on Svalbard, an archipelago under Norwegian jurisdiction, and serves as its administrative hub. Svalbard never had an Indigenous population, and Longyearbyen is today characterized by its transience, young age, and growing international diversity, with approximately 35% hailing from outside Norway. The town underwent considerable socioeconomic transformations, as tourism, the service sector, research, and higher education replaced coal mining as the primary economic sector[100]. Despite its geographical isolation, Longyearbyen boasts modern urban amenities and easy accessibility via its airport.

Longyearbyen had a mean annual air temperature of −3.8 °C during the period of 1990–2018[101]. Alarming climatic shifts have occurred in recent years, with temperatures rising almost 4 °C since meteorological records began in 1899, approximately 3.5 times greater than the global average during the same period[101]. Longyearbyen's permafrost is continuous and warm and was characterized by a mean annual ground temperature at a two-meter depth of −1.4 °C in 2019 (Table 1). Between 2000 and 2019, the permafrost warmed at a rate of 0.12 °C/yr (Table 1). Projections under high emission scenarios suggest that near-surface permafrost in coastal and low-lying areas could thaw before the century's end (Table 1). The town's vulnerability to natural hazards is heightened because of its proximity to steep permafrost slopes. Permafrost thaw could contribute to unstable slopes and an elevated risk of landslides and debris flows, exacerbated by more frequent heavy precipitation events and an increase in winter rainfall[102,103]. Projected increases in temperature and annual precipitation[103] are also expected to cause increased flooding, river and coastal erosion, and a surge in snow and slush avalanches in the coming years.

The Avannaata municipality, created in 2018, encompasses the northwestern regions of Greenland, including four towns and 23 villages with a total of approximately 11,000 residents. Ilulissat, located in the south of the municipality, is the municipal administrative center with approximately 5000 inhabitants[104], whereas Qaanaaq, located in the north, has more than 600 inhabitants[104]. The population is predominantly Indigenous (Kalaallit, Inughuit, and mixed Inuit-Danish) but includes many Danish Greenlanders and international residents, most of whom are transient. Greenland was a Danish colony until 1953 and progressively gained political and economic autonomy with the Home Rule in 1979 before being granted self-government and autonomy in all social spheres in 2009. Ilulissat was originally established in 1741 by a Danish missionary as a trading station under the colonial name Jakobshavn[105]. Currently, the mainstay of its economy is commercial fisheries and tourism, which, together with public services and construction, provide the most employment. Qaanaaq was founded more recently in 1953 as a relocation of Thule, where the United States Air Force constructed an air base[106]. Today, subsistence hunting and fishing, as well as the public and service sectors, are crucial for Qaanaaq's economy. Ilulissat, its adjacent UNESCO World Heritage ice fjord, and the Qaanaaq area are also important hubs for climate and permafrost research.

Ilulissat and Qaanaaq had mean annual air temperatures of −3.7 and −8.9 °C during the periods of 1991–2020 and 1995–2020, respectively[107]. Air temperatures have increased globally at a rate of 0.37 °C/decade over

Greenland from 1961 to 2015[108]. Both settlements are underlain by continuous permafrost, yet the mean annual ground temperatures at a two-meter depth ranged from −2.5 °C in Ilulissat to −7.5 °C in Qaanaaq in 2019 (Table 1). Between 2000 and 2019, the permafrost warmed at a slightly faster rate of 0.08 °C/yr in Ilulissat than in Qaanaaq at 0.06 °C/yr (Table 1). Most of the deglaciated terrains within the municipality are characterized by the presence of bedrock, limiting the threat of coastal erosion. However, owing to the often frost-susceptible nature of the sedimentary deposits in the region[109,110], ground surface deformations induced by permafrost thaw are the most concerning hazardous process, especially because of their impacts on infrastructure and the resulting need for frequent repairs[110]. Although Qaanaaq residents currently experience more serious permafrost-related issues[111], Ilulissat is facing increasing challenges linked to the need for town expansion due to growth in economic activities and a predicted increase in the number of residents, tourists, and foreign employees.

## Main differences between and within communities/study areas

The study regions exhibit large disparities in societal and environmental factors. Societal differences encompass demographic characteristics, economic inequalities, and governance. Longyearbyen stands out as an international university and tourist settler community with strategic Arctic importance. In contrast, Greenland, Canada, and Russia are home to Indigenous communities that are deeply rooted in their cultural traditions. Within the Canadian study area, Inuvik differs greatly from Tuktoyaktuk, Fort McPherson, and Aklavik in terms of infrastructure, Indigenous population proportions, and various economic and societal factors. Inuvik's history is tied to the relocation of residents from Aklavik, whereas Tuktoyaktuk faces coastal erosion challenges and a history closely intertwined with oil and gas extraction. Longyearbyen operates under a Western democratic system, whereas Greenland is moving toward greater self-determination and land ownership. The Canadian study area is characterized by Indigenous self-governance and comanagement systems within a Western nation-state framework. Finally, the Sakha Republic (Yakutiya) operates within yet another distinct regulatory context influenced by Russia's centralized governance and lack of recognition of Indigenous land rightsholders.

All the study regions are underlain by continuous permafrost, but the surface characteristics and subsurface conditions vary. The ground temperatures of Longyearbyen and Greenland are higher but change more slowly than those of Canada and Russia, where temperatures are increasing rapidly (Table 1). Bykovskiy and Tiksi are situated on colder permafrost but experience the highest ground temperature increase rates observable across the Arctic (Table 1). Our study regions consist of coastal and inland communities, each facing unique challenges related to permafrost thaw. Their geological and permafrost characteristics (specific features of frozen ground and ice formations) vary considerably, impacting their susceptibility to erosion and ground instabilities. The topography ranges from mountainous to flat delta landscapes, influencing accessibility, and vulnerability to natural disasters. Distinct landforms and vegetation patterns, such as treeline presence or absence, contribute to the diverse ecological makeup of each region. In summary, our study areas encompass a rich diversity of characteristics, both in terms of human societies and the natural environment, thus representing a wide variety of Arctic contexts.

## Primary data collection

The Nunataryuk project consisted of a consortium of twenty-six research institutes that carried out a comprehensive six-year investigation into rapidly changing permafrost regions in the Northern Hemisphere. The project aimed to answer pressing questions about the role of permafrost thaw in the global climate system and the consequences for ecosystems, the economy, the built environment, and the health of people living in Arctic (near-) coastal regions. Between 2017 and 2023, engineering, physical, environmental, social, and health scientists investigated permafrost thaw-related risks in the study areas (Longyearbyen (Svalbard, Norway), the Avannaata municipality (Greenland), the Beaufort Sea region and the

Mackenzie River Delta (Canada), and the Bulunskiy District in the Sakha Republic (Russia)) (c.f. Fig. 1 and section "Description of the study areas"), which are characterized by the presence of continuous permafrost. The study areas, facing both shared and unique challenges associated with permafrost thaw, were mostly chosen on the basis of long-standing established research relationships. Our transdisciplinary methodological approach[47,48] entailed combining diverse perspectives to ensure that scientific, Indigenous, and other local views were valued and integrated in the sense of a two-eyed seeing approach and multidirectional knowledge exchange[18,112]. This included capacity sharing, cross-cultural collaboration, and knowledge transfer at the policy and science interface[113].

Throughout the study period, from 2017 to 2023, the scientists involved in the project conducted manifold investigations in the four study areas to improve their understanding of the causal relationships among the physical drivers, permafrost thaw hazards, and their societal consequences. In large-scale interdisciplinary projects such as Nunataryuk, quantifying the total volume of data retrieved by various researchers and describing all methods adopted in different study regions pose considerable difficulties. However, an overview of the carried out investigations is provided as follows. Within the physical, environmental and engineering sciences, specific attention was given to permafrost regimes and environmental changes. Geotechnical and geophysical investigations were performed and accompanied by hydrological surveys[114], soil sampling[115], and remote sensing studies[116], which were combined with modeling[117] and mapping[53,118,119]. Within health, social, and engineering sciences, workshops, public meetings, focus groups, a survey[19,35,36], interviews, and informal conversations were held in the different regions to engage in knowledge exchanges with local rightsholders and stakeholders[113]. A detailed overview of these investigations can be found in Supplementary Table 1. Using this mixed methods approach, both researchers' and locals' observations and perceptions were gathered and integrated to obtain a comprehensive and inclusive representation of permafrost thaw risks impacting the different study areas. This information was gathered during the stage of primary data collection in a variety of disciplines, as described above, and provided the qualitative inputs for the risk analysis. The following section delves into the 3-step process of the risk analysis (Fig. 10 and Supplementary Table 2 for additional details), including data collection on risks within the consortium, thematic network analysis (permafrost thaw risk and hazard identification), and the risk ranking process, i.e., consortium ranking and local expert evaluation of framework and risks.

## Risk analysis (Steps 1 to 3)

A wide range of participants provided us with qualitative inputs from the early stages of the project (Primary data collection and Step 1 in Fig. 10), including community residents, hunters, Elders, land users, council members, scientists, Indigenous knowledge and rightsholders, municipal government administrators, urban planners, civic leaders, representatives from various professional sectors (construction, services, energy suppliers, fishing industry, etc.) and other local experts. During this process and until the end of the project, the consortium members maintained a continuous dialog and regularly consulted local stakeholders. Workshops (LEE, local expert evaluation, in Fig. 10) were finally held in Ilulissat (Greenland), Aklavik, Inuvik (Canada), and Longyearbyen (Svalbard, Norway) to present the findings from the risk analysis and gather local expert feedback and evaluations. This community-based participatory research approach[18] was characterized by its responsiveness to local research needs and stakeholder engagement, achieved by collaborating intensely with community members.

## Step 1: Data gathering within the consortium

Between 2018 and 2021, we collaborated with the consortium scientists to consolidate knowledge regarding permafrost thaw risks obtained from their primary data collection in the study areas (c.f. section "Primary data collection" and Supplementary Table 1). Qualitative data pertaining to the physical processes linked to permafrost thaw, associated hazards, and resulting societal consequences for local communities were synthesized

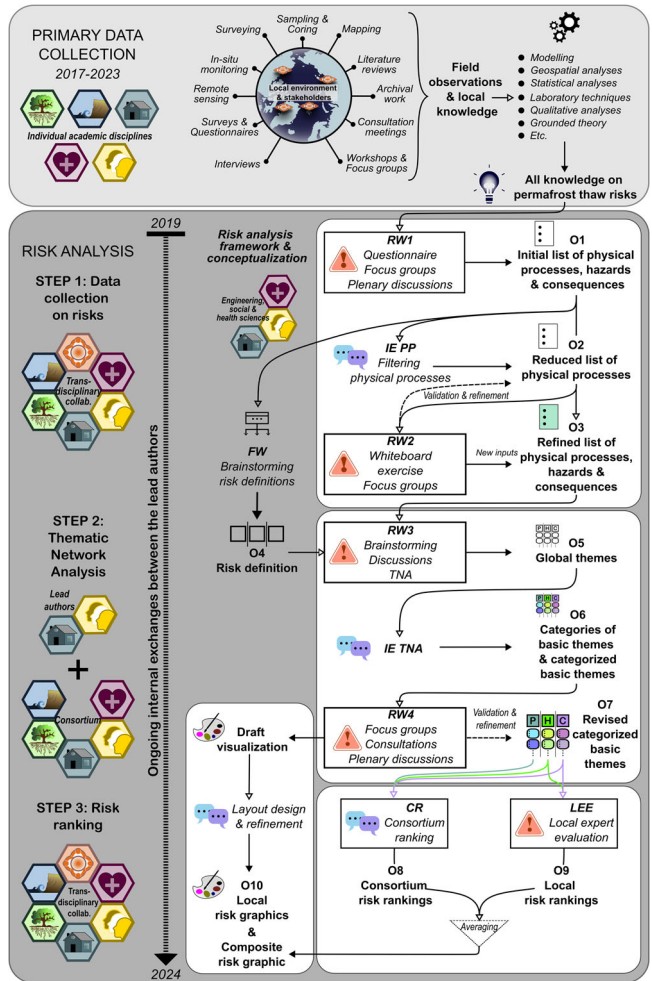

**Fig. 10 | Permafrost risk analysis workflow diagram.** The diagram depicts the inputs, data collection, analysis steps, and outputs of both the primary data collection and risk analysis. RW and FW stand for risk workshop and framework workshop, respectively. IE refers to internal exchanges, whereas PP and TNA refer to physical processes and thematic network analysis, respectively. O1 stands for Output1, etc. The orange hexagon represents local rightsholders' and stakeholders' knowledge. The purple, yellow, petrol blue, green, and light blue hexagons symbolize health, social, engineering, environmental, and physical sciences, respectively.

through a series of workshops and discussions (Step 1 in Fig. 10). One workshop (FW in Fig. 10), conducted in 2019, was focused on developing a comprehensive framework[39] and defining the nature of risk as encompassing multiple dimensions (e.g., physical and social) and perceptions (O4 in Fig. 10). Two risk workshops (RW1 and RW2 in Fig. 10) were conducted during the general assemblies of the Nunataryuk project. During the first workshop held in 2019, participants were organized into regional break-out groups according to their specialization and tasked with defining risks and associated uncertainties, as well as identifying permafrost thaw-related concerns for humans (O1 in Fig. 10). The insights provided were then discussed in groups corresponding to the different study areas and within a panel discussion involving scientists and local (Indigenous) stakeholders. The second risk workshop was conducted online in 2021, where once again, participants were grouped according to their specialization. At this stage, the consortium scientists further refined and expanded upon the compiled examples to enumerate additional (direct and indirect) risks, physical processes, and consequences associated with permafrost thaw (O2 and O3 in Fig. 10). This two-year process resulted in the creation of a comprehensive inventory (O3 in Fig. 10), which was used as input in our risk analysis.

On the basis of this inventory, we identified and ranked key risks from permafrost thaw in further collaboration with project scientists and local experts.

## Step 2: Thematic network analysis—permafrost thaw risks and hazard identification

The data collected in Step 1 were analyzed through a thematic network approach and thus categorized on the basis of different risk components (Step 2 in Fig. 10). Thematic network analysis is a systematic method for organizing qualitative data, aiming to uncover underlying structures and to depict different orders of themes. Following Attride-Stirling[56], the method consisted of systematically extracting the following:

- lowest-order premises evident in texts (or basic themes), which in our case corresponded to the examples of permafrost thaw-related physical processes, hazardous events, and consequences provided by the scientists and local stakeholders in Step 1;
- the categories of basic themes grouped together, which in our case were groups of physical processes, key hazards, and groups of societal consequences (life domains); and
- the superordinate or global themes, which in our case were finally identified as the main components of risk, namely, the permafrost thaw-related physical processes, hazards, and societal consequences.

These global themes (O5 in Fig. 10) were collaboratively determined during a third risk workshop involving social, health, and engineering scientists (RW3 in Fig. 10). Through a series of internal exchanges (IE TNA in Fig. 10), the thematic network analysis then led to the identification of categories of basic themes, including the formulation of the key hazards and categorization of the inputs gathered in Step 1 per group of physical processes and societal consequences (O6 in Fig. 10 and Table 2). The thematic network analysis finally underwent validation within the consortium through a last risk workshop (RW4 resulting in O7 in Fig. 10) and other internal exchanges. This systematic and iterative process engaged the consortium scientists in scrutinizing the results by examining and refining the groups of physical processes, the key hazards, and the groups of consequences, their definitions and basic themes.

## Step 3: Risk ranking

Following the validation of the thematic network analysis, we ranked the importance of (i) the groups of physical processes triggering the occurrence of the key hazards and (ii) the occurrence of the key hazards causing consequences for life domains (CR, consortium ranking, in Fig. 10). Subsequently, the outcomes of these deliberations were returned to the collaborating Arctic communities, who were invited to provide their own rankings of the key hazards and consequences, thus enabling a reciprocal interpretation of the results (LEE, local expert evaluation, in Fig. 10).

First, the relationships between the groups of physical processes and key hazards were ranked by physical, environmental and engineering scientists as a synthesized perspective across all study areas (i.e., same score for all study areas). Intradisciplinary expertise was specifically sought to verify the individual rankings of the physical processes (i.e., hydrologists verifying the weights given to hydrological cycles on given key hazards, etc.). Second, the expertise of social, health, and engineering scientists was used to rank the relationships between the key hazards and impacted life domains. In contrast with the physical processes, the importance of the consequences for life domains was assessed per study area by deliberating with scientists who had conducted studies in the respective communities. Values ranging from 0 to 2 were used as integer scores to rank the relationships between the groups of physical processes, key hazards, and groups of societal consequences. A value of 0 was given to the relationships assessed as irrelevant or least relevant, whereas a value of 2 was attributed to the most relevant relationships (O8 in Fig. 10).

The results of the risk analysis undertaken by the consortium were then shared with Arctic communities in early 2023. In particular, four workshops

**Table 2 | Global themes, categories of basic themes, and basic themes identified from the data via thematic network analysis**

| Global themes | Categories of basic themes | Basic themes |
|---|---|---|
| PHYSICAL PROCESSES: Climatic, environmental, and anthropogenic drivers of and processes resulting from permafrost thaw. | Changes in climate & weather conditions: Rising temperatures and changing precipitation patterns cause changes in surface energy balance. | • Increase in air and water temperature |
| | | • Rising frequency and magnitude of extreme weather events |
| | | • Changes in freeze/thaw cycles |
| | | • Decrease in sea ice coverage |
| | | • Increased winter-precipitation |
| | Human influences: Human activities cause climate change and impact ecosystems, through the emission of greenhouse gases and the release of contaminants. | • Greenhouse gas emissions |
| | | • Land disturbances and landscape modifications (inadequate construction practices, river/water flow diversions, etc.) |
| | | • Leaks and contaminations (oil spill, burying of radioactive waste, etc.) |
| | | • Increase in tourism (pressure on resources, need for infrastructures, etc.) |
| | Terrestrial & subsea permafrost degradation: Permafrost degradation releases carbon into the atmosphere, and manifests through rising ground temperatures, a deeper active layer, melting ground ice and changes in soil and sediment properties. | • Rising ground temperatures |
| | | • Changes in active layer thickness |
| | | • Thawing of ground ice leading to thermokarst phenomena |
| | | • Changes in soil properties (pore water pressure, bearing capacity, etc.) |
| | Ground instabilities & erosion: Ground instabilities caused by permafrost thaw and extreme weather events, such as landslides, thermokarst (lakes), erosion, and rockfalls, impact terrestrial, fluvial, and coastal ecosystems. | • Ground surface deformations (frost heave and differential thaw settlements) |
| | | • Active layer detachments, retrogressive thaw slumps, landslides, rockfalls |
| | | • Gullying |
| | | • Solifluction/gelifluction |
| | | • Changes in sedimentation/erosion patterns |
| | | • Increase in coastal and fluvial erosion rates |
| | | • Increase in turbidity of coastal waters |
| | Changes in flora & fauna: Permafrost landscape transformation alters the life cycles, habitats, and biodiversity of living organisms, adding pressure on food webs and ecosystem services. | • Fragmentation and loss of habitats |
| | | • Loss of biodiversity (species evolution, vegetation composition, invasive species, etc.) and changes in food webs, communities, and species-interactions |
| | | • Changes in (evapo-)transpiration (surface energy balance, etc.) and vegetation patterns |
| | | • Changes in gross productivity of the ecosystem |
| | | • Arctic browning/greening (e.g., shrubification) |
| | | • Changes in animal migration patterns and routes |
| | | • Changes in organism metabolism/activity |
| | | • Increased microbial decomposition of stored organic matter |
| | | • Accumulation of contaminants in living organisms |
| | Hydrological changes: Changes in water cycle and (sub-)surface hydrology are largely mediated by the prevailing status of permafrost and ground ice conditions Permafrost and ground ice conditions affect soil moisture and the water cycle. In turn, snow cover, soil moisture, and ground-water flows impact the state of permafrost and ecosystems. | • Changes in snow properties and cover extent and duration |
| | | • Changes in water runoff/drainage regimes |
| | | • Changes in evaporation, transpiration, and precipitation patterns |
| | | • Changes in surface- and ground-water regimes and soil moisture |
| | Changes in biogeochemical cycles: Complex interactions between thawing permafrost soils, microorganisms, vegetation, and the limnic and oceanic realm determine the fate of ecosystems under climate change. | • Changes in biogeochemical cycles (carbon, nitrogen, phosphorus fluxes) |
| | | • Release of organic matter and nutrients |
| | | • Release of contaminants and heavy metals (Persistent Organic Pollutants, mercury, chemicals of emerging Arctic concern, etc.) |
| | | • Spread of infectious diseases |
| | | • Ocean acidification, increased benthic alkalinity, changes in salt fluxes |
| | | • Arctic browning/greening |

**Table 2 (continued) | Global themes, categories of basic themes, and basic themes identified from the data via thematic network analysis**

| Global themes | Categories of basic themes | Basic themes |
|---|---|---|
| HAZARDS: Harmful phenomena or events with adverse impacts on humans, material assets, livelihoods, and ecosystems. | Infrastructure failure: Permafrost thaw and associated erosion endanger housing, communication, transport, and energy infrastructures upon which Arctic communities depend. | • Destruction or destabilization of infrastructures (collapsing, subsiding, etc.) |
| | | • Damages to residential buildings (slanted floors and ceilings, doors and windows not closing, etc.) |
| | | • Damages to industrial infrastructure, power/fuel supply facilities, refuse and burying sites (pipes, mining sites, offshore platforms, landfills, etc.) |
| | | • Damages to and loss of cultural heritage and thawing graves/cemeteries |
| | | • Damages to communication infrastructure (antennas, towers, etc.) |
| | Disruption of mobility & supplies: Thawing permafrost damages transportation infrastructure, food and drinking water supply facilities. Extreme weather and erosion also disrupt navigation routes, limiting access to resources. | • Destruction/disruption of navigation routes (changes in navigational river channels due to sedimentation/silting, etc.) |
| | | • Destruction/disruption of transportation infrastructure such as airstrips, roads (incl. ice roads), harbors, and tracks (potholes, landslides, flooding, etc.) |
| | | • Damage to water/food supply facilities (pipes, ice cellars, and caches, etc.) |
| | Decrease in water quality: The release of organic carbon, nutrients, sediments, and contaminants into aquatic systems deteriorates the water quality and affects ecosystems, food security, and access to clean water. | • Ocean acidification can lead to biodiversity loss and loss of habitat (e.g., anadromous fish) |
| | | • Eutrophication and consequent water anoxia |
| | | • Changes in pollutant concentration |
| | | • Higher concentration of mercury in rivers and ocean |
| | | • Increase in water stratification hampering vertical mixing |
| | Challenges for food security: Biodiversity loss, habitat destruction, and declining animal populations pose challenges for subsistence activities. Landscape transformations and infrastructure failures can lead to the release of contaminants and disrupted travel routes, compromising food supply and quality. | • Changes in species migration routes (reindeer, muskox..), distribution (caribou), and vegetation composition (shrubification, etc.) |
| | | • Loss of biodiversity, habitats, and population size |
| | | • Damages to ice cellars and caches |
| | | • Contaminant fluxes from decommissioned oil and gas wells |
| | | • Reduced/disrupted access to fishing/hunting/berry and medicine picking grounds |
| | Exposure to infectious diseases & contaminants: Thawing permafrost and erosion contribute to the diffusion of mercury, the spread of infectious diseases, and trigger the development of harmful algae blooms, endangering aquatic life. Unsecured hazardous waste may also release contaminants. | • Transformation, release of, and exposure to environmental contaminants, such as mercury |
| | | • Diffusion or increased diffusion of infectious existing and ancient diseases caused by viruses and bacteria (anthrax, etc.) |
| | | • Potential increase in climate-sensitive diseases such as tick-borne diseases, tularemia, anthrax, and vibriosis |
| | | • Development of harmful algae blooms |
| | | • Eroding cemeteries and thawing graveyards leading to the reappearance of and exposure to plague and other viruses |
| | | • Lack of protective measures, f.e. against thawing of landfills leading to exposure to environmental contaminants |
| | | • Challenges occurring for fauna immune systems |
| SOCIETAL CONSEQUENCES: Direct and indirect consequences impacting various aspects of human life, including their ecosystems. | Culture & language: Increased permafrost thaw leads to significant impacts on livelihoods and subsistence, as well as heritage and identity. For instance, changes in food sources can disrupt traditional ways of life, and there may be a threat to the transfer of intergenerational knowledge. | • Concern about the impact on livelihood and subsistence |
| | | • Concern about loss and/or maintenance of cultural, tangible and intangible heritage |
| | | • Concern about impacts on culture, identity, language, social fragmentation |
| | | • Decreased reliability of ice cellars |
| | | • Concern about over-reliance on and excessive use of new technology and consequent loss of land-based knowledge transfer |

**Table 2 (continued) | Global themes, categories of basic themes, and basic themes identified from the data via thematic network analysis**

| Global themes | Categories of basic themes | Basic themes |
|---|---|---|
| | Health & well-being: The physical and mental health and well-being of communities are affected by permafrost thaw. This includes a higher potential for injuries due to changing landscapes and greater uncertainty regarding safe food consumption. | • Impact on physical health and safety (e.g., injuries, loss of lives) |
| | | • Concern for resulting technological hazards (industrial pollution, fires, chemical and oil spills, soil contamination by toxic wastes, garbage dumps, landfills, etc.) |
| | | • Concern about potential contamination and thus loss of fishing, hunting, trapping, and gathering grounds; insecurity about the safety of harvest and consumption of country foods |
| | | • Concern for greater dependence on external food sources |
| | | • Impacts on mental health due to uncertainty regarding impacts on freshwater supply and country food, as well as economic hardship |
| | | • Concern about diseases spreading from thawing animal graves in coastal communities and associated mental health impacts |
| | | • Direct or indirect exposure of humans and animals to infectious diseases and contaminants through food and water sources, and associated risks of neurobehavioral, reproductive, cardiovascular, endocrine, and carcinogenic effects |
| | | • Decreased quality of indoor environment |
| | Costs & economy: Permafrost thaw necessitates repairs, investments in new equipment, and the adoption of physical protective measures. The communities, families, and individuals bear the financial burden of these necessary adaptation measures. Additionally, there is an increased reliance on store-bought food, which can impact local economies. | • Increased costs, through the need for new equipment or repairs |
| | | • Economic challenges and financial losses (e.g., higher costs for external food supplies, electricity, materials, and costs of repairs, emergency responses, decommission, protective measures, etc.), potentially resulting in economic hardship |
| | | • Concern about increased costs and decreasing benefits of subsistence fishing, hunting, and gathering (e.g., more time and resource-intensive hunting and fishing or decreasing reindeer pastures, etc.) |
| | | • Diseases and health care leading to high costs for individuals, families, and communities |
| | | • Disruption or loss of serviceability affecting economic activities and benefits |
| | Ecosystem: Disruptions and changes caused by permafrost thaw, such as altered nutrient fluxes, bioaccumulation of mercury, and outbreaks of anthrax disease, raise concerns about the normal functioning of ecosystems. | • Decreased health of flora and fauna, loss of biodiversity, food web changes |
| | | • Bioaccumulation of diseases and contaminants such as mercury harming food sources and human consumption |
| | | • Anthrax disease affects animals and herds, which are important means of sustenance |
| | | • Higher dependence on store-bought/imported foods may cause an increase in the carbon footprint of local food consumption |
| | Recreation & being in nature: Recreational activities and the simple act of being in nature are adversely affected by permafrost thaw. Changes in navigable water and terrestrial paths, as well as more challenging access to camps, hinder recreational opportunities and the ability to connect with nature. | • Harder to walk across land in wetter tundra (e.g., during berry picking, hunting, etc.) |
| | | • More difficult access to camps through travel routes (e.g., boats getting stuck on sand bars resulting from erosion and modification of sedimentation patterns, etc.) |
| | | • Concern about safety while being in nature and engaging in recreational activities |
| | Planning & fate control: Permafrost thaw impacts planning and the sense of control over one's fate. The potential loss of connectivity and supply irregularities pose significant challenges. Communities may need to explore new water and food sources to adapt to the changing environment. | • Uncertainty regarding future investment, potential loss of home, or need for relocation |
| | | • Adaptation measures to changing environment and climate needed in local, municipal, and regional planning |
| | | • Challenges regarding recruitment, retainment, and education of qualified staff |

**Table 2 (continued) | Global themes, categories of basic themes, and basic themes identified from the data via thematic network analysis**

| Global themes | Categories of basic themes | Basic themes |
|---|---|---|
| | | • Concern for depopulation/out-migration |
| | | • Concern for disrupted mobility and loss of accessibility |
| | | • of connectivity between and within settlements and camps (e.g., closed roads, unnavigable rivers, etc.) |
| | | • Possible disruption of chains of supply and irregularity of supply |
| | | • Need for new food and water sources may arise |
| | | • Concern about a decrease in country food variability |

The list of basic themes is not exhaustive.

were organized in Ilulissat (Greenland), Aklavik, Inuvik (Canada), and Longyearbyen (Svalbard, Norway), during which local rightsholders and stakeholders were given the opportunity to rank the importance of the identified hazards and consequences for life domains within their community[57]. Approximately forty participants took part in the local expert evaluation workshop in Ilulissat, forty-two in Inuvik and Aklavik, and eleven in Longyearbyen[57]. To this end, the participants first evaluated the relevance of each key hazard as low or high by using green (low relevance) and red stickers (high relevance). Second, the consequences of the hazards on life domains, which were highlighted as pertinent, were evaluated as slightly, moderately, or very relevant using an unlimited number of corresponding green, yellow, and red stickers. Open discussions subsequently enabled the participants to express their perceptions of the local impacts of permafrost thaw and the challenges in managing permafrost thaw risks. Owing to the war between Russia and Ukraine, it was not possible to plan local expert evaluation workshops in the Bulunskiy District. This gap was partially mitigated through collaboration with outside researchers who had extensive prior experience working with Russian permafrost communities, thereby enhancing the robustness of our evaluation.

The risk workshop in Greenland took place in Ilulissat in January 2023 and consisted of individual meetings with stakeholders from different professional and ethnic backgrounds and an open community meeting. The workshop specifically focused on validating the identified key risks within the Avannaata municipality and gathering further local perceptions of risks. This first workshop enabled us to test and adapt the methodology in the other study areas. Within the Beaufort Sea region and the Mackenzie River Delta, several workshops and individual meetings were held in January and February 2023. A risk workshop was conducted in Inuvik with professionals representing Indigenous governments, local permafrost researchers, and local and regional organizations. A similar approach to the Ilulissat workshop was adopted to enable the participants to rank key hazards and consequences with stickers of different colors on the basis of their relevance. A workshop was then held in Aklavik, involving primarily members of the Aklavik Hunters and Trappers Committee. The Aklavik workshop group expressed greater concerns regarding all key hazards. For this reason, we adapted the methodology and distributed an unlimited number of green and red stickers to rank the importance of the key hazards. Thereafter, the number of green, yellow, and red stickers remained unlimited to assess the importance of the consequences for life domains. On Svalbard, the risk workshop was held in Longyearbyen in March 2023, again bringing together early career and senior researchers from diverse disciplines, including environmental, engineering, health, and social sciences, working on various permafrost projects and representatives from local authorities. During the risk workshops, key permafrost thaw hazards and their consequences for life domains were discussed and ranked. In addition, at all three study sites, the state of permafrost research, indicators of permafrost thaw impacts[120], and ways forward were addressed in dedicated sessions.

The rankings derived during each of the community risk workshops with colored stickers (referred to as local expert rankings) were translated to integer scores via a scale similar to that of the consortium (O9 in Fig. 10).

The green, yellow, and red stickers were assigned values of 0 for irrelevance or lowest relevance, 1 for moderate relevance, and 2 for highest relevance, respectively. Therefore, key hazards that had received only green stickers from participants were considered irrelevant in the study area in question. The relationships between these hazards and the six groups of consequences (life domains) were accordingly given scores of 0. For the key hazards that received red stickers (signifying their importance in the study area), local rankings of the consequences were then computed by summing the integer scores associated with the sticker colors and dividing the sum by the total number of stickers. For example, if the consequences of infrastructure failure on the life domain costs and economy received two yellow and three red stickers within a study area, the final ranking of the relationship between infrastructure failure and costs and economy would be equal to 1.6 (i.e., $(2 * 1 + 3 * 2)/5$).

To obtain the final rankings representing the importance of the relationships between the key hazards and impacted life domains, the local expert rankings were then averaged together with the consortium rankings for each hazard-life domain connection and per study area. By averaging the rankings, we gave equal weight to both scientific and local perceptions, thereby considering them as a spectrum rather than as entirely distinct viewpoints. In the case of the Beaufort Sea/Delta region specifically, the scores gathered during both the Inuvik and Aklavik workshops were averaged together with the consortium rankings. In the case of the Bulunskiy District, only the consortium rankings were available to characterize the severity of permafrost thaw consequences.

The permafrost thaw risk levels resulting from our analysis were represented as risk graphics (O10 in Fig. 10). Local risk graphics (Figs. 6 to 9) were generated per region, allowing us to compare the study areas. In addition, a risk graphic (Fig. 2) providing a composite overview across the four study areas was produced by averaging the rankings of all the sites. This synthesis of the different study areas is where the novelty of our research resides, identifying key risks of permafrost thaw, not only in sectoral case studies but also across disciplines, stakeholder perceptions, and permafrost localities. The connections between the groups of physical processes and key hazards represented on the composite risk graphic were derived from the consortium rankings. In contrast, the connections between the key hazards and the groups of consequences (life domains) represented in the local and composite risk graphics were calculated by averaging the consortium and local expert rankings, as explained previously. The connections were finally visualized as follows in all the risk graphics: ranking <0.5: not represented, [0.5–1]: least relevant, [1–1.5]: somewhat relevant, [1.5–2]: very relevant.

## Methodological challenges

The iterative methodological process we have described underscores the collaborative approach to risk analysis and the integration of a wide range of knowledge sources. Moreover, during our study, we encountered several limitations and challenges that influenced the organization and outcomes of our risk analysis, and that will necessitate careful consideration in future research efforts. External factors, such as the Russian war on Ukraine and the global COVID-19 pandemic, introduced additional complexities to our

research environment. Notable challenges also arose from the need to integrate different scientific disciplines and their methodologies, as well as the diverse perceptions of risk among various stakeholders. Importantly, risk definitions and perceptions can vary significantly among scientific disciplines, individuals, and communities. These variations are influenced by a wide array of factors, including the level of transience, place attachment, cultural background, livelihood, occupation, educational level, socio-economic status, governance structure, and personal beliefs (including religious and political views). Furthermore, value systems, the knowledge base, knowledge sharing, and the degree to which different types of knowledge are integrated influence risk perception. Within the scientific community, for example, risk perception is also influenced by individuals' scientific backgrounds and areas of specialization. Locally, it is often challenging to single out permafrost thaw-induced challenges from broader issues associated with climate change within community perceptions. Similarly, isolating the impacts of permafrost thaw from other anthropogenic and environmental factors remains difficult. Permafrost risks result from multidirectional and multifactorial relationships, and local observers often do not distinguish between different causes and impacts of changes. These diverse elements, coupled with local complexities in permafrost and geological characteristics, play pivotal roles in shaping how risks are perceived and how communities respond to them. Finally, although we addressed some of the main factors, we did not conduct an in-depth analytical comparison identifying what causes the differences between the case studies. This is a possible next step to be taken in future research.

## Data availability

The results of the thematic network analysis and risk ranking are openly available on Zenodo.org at https://doi.org/10.5281/zenodo.14173601.

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

## Acknowledgements
This research has received funding from the European Union's Horizon 2020 research and innovation program under grant agreement No. 773421 (Nunataryuk) and from the EU Horizon Europe, grant agreement No. 101133587 (ILLUQ). Parts of the research were conducted under UArctic funding from the Ministry of Higher Education and Science in Denmark and supported by InfraNorth (European Research Council Advanced Grant project, grant agreement No. 8856646) and the Academy of Finland (grant No. 356604). We would like to extend our deepest gratitude to all the contributors to this work over the years, including local communities, experts, knowledge holders, our Nunataryuk colleagues, and affiliated researchers. Special thanks to the Aklavik Hunters' and Trappers Association, the Gwich'in Tribal Council, the Aurora Research Institute, Avannaata municipality, Nukissiorfiit, Longyearbyen Lokalstyre, the University Center in Svalbard, as well as the North-Eastern Federal University in Yakutsk, the Institute for Humanities Research and Indigenous Issues, and the Permafrost Institute n.a. Melnikov of the Russian Academy of Sciences. We would also like to express our appreciation to Grid Arendal, as well as Sebastian Laboor (Alfred Wegener Institute) for his valuable assistance in mapping the study areas, and to our reviewers for their insightful feedback and suggestions.

## Author contributions
Susanna Gartler, Johanna Scheer, and Alexandra Meyer organized and supervised the data collection for the risk analysis through internal consortium workshops and follow-up meetings. Susanna Gartler, Johanna Scheer, Alexandra Meyer, Khaled Abass, Annett Bartsch, Natalia Doloisio, Jade Falardeau, Gustaf Hugelius, Anna Irrgang, Jón Haukur Ingimundarson, Leneisja Jungsberg, Hugues Lantuit, Joan Nymand Larsen, Rachele Lodi, Victoria Sophie Martin, Louise Mercer, David Nielsen, Paul Overduin, Olga Povoroznyuk, Arja Rautio, Peter Schweitzer, Niek Jesse Speetjen, Soňa Tomaškovičová, Ulla Timlin, Jean-Paul Vanderlinden, Jorien Vonk, Levi Westerveld, Thomas Ingeman-Nielsen, as well as other members of the Nunataryuk consortium, contributed with their data and knowledge throughout this process. Susanna Gartler, Johanna Scheer, Alexandra Meyer, Leneisja Jungsberg, Joan Nymand Larsen, Arja Rautio, Peter Schweitzer, Ulla Timlin, and Thomas Ingeman-Nielsen acquired the funding for the risk workshops. Susanna Gartler, Alexandra Meyer, Jón Haukur Ingimundarson, Leneisja Jungsberg, Joan Nymand Larsen, Paul Overduin, Arja Rautio, Peter Schweitzer, Ulla Timlin, Thomas Ingeman-Nielsen, and others organized the workshops in the study areas. Susanna Gartler, Johanna Scheer, and Alexandra Meyer performed the data processing and risk analysis and computed the risk rankings. Levi Westerveld designed the risk graphics, which were adapted to the data by Johanna Scheer. Susanna Gartler and Johanna Scheer produced the other figures. Susanna Gartler, Johanna Scheer, and Alexandra Meyer wrote the initial manuscript draft and the final manuscript and were primarily responsible for writing, editing, and revising. Susanna Gartler, Johanna Scheer, Alexandra Meyer, Khaled Abass, Annett Bartsch, Natalia Doloisio, Jade Falardeau, Gustaf Hugelius, Anna Irrgang, Jón Haukur Ingimundarson, Leneisja Jungsberg, Hugues Lantuit, Joan Nymand Larsen, Rachele Lodi, Victoria Sophie Martin, Louise Mercer, David Nielsen, Paul Overduin, Olga Povoroznyuk, Arja Rautio, Peter Schweitzer, Niek Jesse Speetjen, Soňa Tomaškovičová, Ulla Timlin, Jean-Paul Vanderlinden, Jorien Vonk, Levi Westerveld, Thomas Ingeman-Nielsen supported the writing, editing and revision process.

## Funding

## Competing interests
The authors declare no competing interests.

## Additional information

[1]Department of Social and Cultural Anthropology, University of Vienna, Vienna, Austria. [2]Austrian Polar Research Institute (APRI), Vienna, Austria. [3]Department of Ecology and Environmental Science, Umeå University, Umeå, Sweden. [4]Department of Environmental & Resource Engineering, Technical University of Denmark, Kongens Lyngby, Denmark. [5]Department of Environmental Health Sciences, College of Health Sciences, University of Sharjah, Sharjah, United Arab Emirates.

[6]Research Unit of Biomedicine and Internal Medicine, University of Oulu, Oulu, Finland. [7]b.geos, Korneuburg, Austria. [8]CEARC Research Center, Université Paris Saclay, Orsay, France. [9]Geotop Research Center in Earth System Dynamics, Département des Sciences de la Terre et de l'Atmosphère, Université du Québec à Montréal, Montréal, QC, Canada. [10] Department of Physical Geography and Bolin Centre for Climate Research, Stockholm University, Stockholm, Sweden. [11]Permafrost Research Section, Alfred Wegener Institute Helmholtz Centre for Polar and Marine Research, Potsdam, Germany. [12]Stefansson Arctic Institute and University of Akureyri, Akureyri, Iceland. [13]Nordregio, Stockholm, Sweden. [14]Institute of Geosciences, University of Potsdam, Potsdam, Germany. [15]Institute of Polar Sciences, National Research Council, Venice, Italy. [16]Department of Environmental Sciences, Informatics and Statistics, Ca' Foscari University of Venice, Venice, Italy. [17]Centre for Microbiology and Environmental Systems Science, University of Vienna, Vienna, Austria. [18]Department of Geography and Environmental Sciences, Northumbria University, Newcastle, UK. [19]Max Planck Institute for Meteorology, Hamburg, Germany. [20]Center for Earth System Research and Sustainability, University of Hamburg, Hamburg, Germany. [21]Department of Earth Sciences, Vrije Universiteit Amsterdam, Amsterdam, The Netherlands. [22]School of Environmental Studies, University of Victoria, Victoria, BC, Canada. [23]Biomedicine and Internal Medicine, Faculty of Medicine, University of Oulu, Oulu, Finland. [24]Laboratoire CEARC, Paris Saclay University, Université de Versailles Saint-Quentin-en-Yvelines, Guyancourt, France. [25]Centre for the Study of the Sciences and the Humanities, University of Bergen, Bergen, Norway. [26]Grid Arendal, Arendal, Norway. [27]These authors contributed equally: Susanna Gartler, Johanna Scheer, Alexandra Meyer.
✉e-mail: susanna.gartler@gmail.com; johanna.scheer@protonmail.com; alexandra.meyer@univie.ac.at

