## [Transparent Peer Review file · Communications Earth & Environment]

A transdisciplinary, comparative analysis reveals key risks from Arctic permafrost thaw

Corresponding Author: Dr Johanna Scheer

Version 0:

Decision Letter:

Dear Dr Scheer,

Your manuscript titled "Local risks from Arctic permafrost thaw – A transdisciplinary, comparative analysis" has now been seen by 2 reviewers, and we include their comments at the end of this message. They find your work of interest, but some important points are raised. We are interested in the possibility of publishing your study in Communications Earth & Environment, but would like to consider your responses to these concerns and assess a revised manuscript before we make a final decision on publication.

We therefore invite you to revise and resubmit your manuscript, along with a point-by-point response that takes into account the points raised. Please highlight all changes in the manuscript text file.

In particular, for publication in Communications Earth and Environment, we request that you a) provide novel insights into the impacts of permafrost thaw on the environment and the socio-economic, health, and cultural well-being of local communities in four Arctic regions and b) outline methodological details in the main text, including the definition of key terms and information about survey and workshop participants and (c) discuss the value and the importance of comparisons of different regions.

Please note that your manuscript falls into the following fields:

- Behavioural and social science

Hence, we request that you provide an updated and completed version of our Reporting Summary and upload it with the revised manuscript.

<https://www.nature.com/documents/nr-reporting-summary.zip>

Please use the following link to submit your revised manuscript, point-by-point response to the referees' comments (which should be in a separate document to any cover letter), a tracked-changes version of the manuscript (as a PDF file) and the completed checklist:

Link Redacted

We hope to receive your revised paper within six weeks; please let us know if you aren't able to submit it within this time so that we can discuss how best to proceed. If we don't hear from you, and the revision process takes significantly longer, we may close your file. In this event, we will still be happy to reconsider your paper at a later date, as long as nothing similar has been accepted for publication at Communications Earth & Environment or published elsewhere in the meantime.

Please do not hesitate to contact us if you have any questions or would like to discuss these revisions further. We look forward to seeing the revised manuscript and thank you for the opportunity to review your work.

Best regards,

Martina Grecequet, PhD
Associate Editor,
Communications Earth & Environment
@CommsEarth

EDITORIAL POLICIES AND FORMATTING

Editorial Policy: [Policy requirements](https://www.nature.com/documents/nr-editorial-policy-checklist.pdf) (Download the link to your computer as a PDF.)

- Behavioural and social science
- Ecological, evolutionary & environmental sciences
- Life sciences

<https://www.nature.com/documents/nr-reporting-summary.zip>

Furthermore, please align your manuscript with our format requirements, which are summarized on the following checklist: [Communications Earth & Environment formatting checklist](https://www.nature.com/documents/commsj-phys-style-formatting-checklist-article.pdf)

and also in our style and formatting guide [Communications Earth & Environment formatting guide](https://www.nature.com/documents/commsj-phys-style-formatting-guide-accept.pdf) .

***** DATA:** Communications Earth & Environment endorses the principles of the Enabling FAIR data project (<http://www.copdess.org/enabling-fair-data-project/>). We ask authors to make the data that support their conclusions available in permanent, publically accessible data repositories. (Please contact the editor if you are unable to make your data available).

All Communications Earth & Environment manuscripts must include a section titled "Data Availability" at the end of the Methods section or main text (if no Methods). More information on this policy, is available at <http://www.nature.com/authors/policies/data/data-availability-statements-data-citations.pdf>.

If a community resource is unavailable, data can be submitted to generalist repositories such as [figshare](https://figshare.com/) or [Dryad Digital Repository](http://datadryad.org/). Please provide a unique identifier for the data (for example a DOI or a permanent URL) in the data availability statement, if possible. If the repository does not provide identifiers, we encourage authors to supply the search terms that will return the data. For data that have been obtained from publically available sources, please provide a URL and the specific data product name in the data availability statement. Data with a DOI should be further cited in the methods reference section.

REVIEWER COMMENTS:

Reviewer #1 (Remarks to the Author):

Main Article

General Comments

"Local risks from Arctic permafrost thaw – A transdisciplinary, comparative analysis", describes a study of the impacts of permafrost thaw on the environment and the socio-economic, health, and cultural well-being of local rights- and stakeholders in four different Arctic regions (western Canada, Greenland, Svalbard, and Russia). The risk analysis conducted in this study is novel as it considers both physical processes and societal concerns and draws understanding from diverse disciplines and perspectives via multidirectional knowledge exchange processes.

As the authors point out, risks related to permafrost thawing are not perceived or understood uniformly, neither across scientific disciplines nor by local stakeholders, and they therefore prioritized developing a comprehensive and transdisciplinary understanding of both the environmental and societal implications of permafrost thaw. Their analysis reveals five main risks (as well as their causes and consequences) from permafrost thaw, including infrastructure failure, disruptions of mobility and supplies, decrease in water quality, challenges for food security, and exposure to infectious diseases and contaminants.

This study is truly transdisciplinary, combining knowledge and perspectives from a wide variety of experts, both local and non-local, from start to finish. The study regions/communities were specifically chosen according to long-standing, established research relationships, which likely increases participation and adds value.

Local expert and stakeholder knowledge and perspectives have been meaningfully included and presented in the analysis of the risks associated with permafrost thaw – for example, the relevance of the connections between the key hazards and groups of consequences were calculated by averaging the consortium (scientists') and local expert rankings and are clearly indicated in Figure 2 and in Extended Data figures 1-4 by lines of varying thicknesses.

In my specific page #, line # comments below, I recommend that the author(s) provide up-front, clear definitions of important terms in this manuscript, including, risk, hazard, risk assessment, and consequences (while acknowledging the sentence on page 18, line 463 that briefly refers to the fact that risk definitions (and perceptions) can vary significantly). I also suggest that some additional information about the surveys and workshop participation (within individual communities and comparatively between the study sites), is needed. I looked up reference #76 and see that it addresses both of these particular concerns quite well. Reference #76 (a grant report/deliverable) overlaps with a lot of what is written in the manuscript being reviewed here, but further defines some of the critically important terms such as risk, hazard, etc., and describes the methodology (workshops, surveys, etc.) and results (participation, etc.) in greater detail. My point here is that I am not sure to what degree an article published in a journal such as *Communications Earth & Environment*, with its broad appeal, should require its readers to do so much background reading and/or reading ahead to the Extended Data section to understand some of the key working definitions, figures, and methods presented in the main text. I do feel that clarification of the important terminology, early on in the manuscript, as well as better descriptions of data acquisition methods would help readers better understand the methods and main results of this study and therefore make a bigger impact.

I also feel that this manuscript would make a bigger impact if it were to explicitly state the value in comparing results from four very different Arctic regions (i.e. a pan-Arctic analysis). For example, from page 3, lines 66-68: "Such strategies are essential for addressing the challenges posed by permafrost degradation while considering both the unique and shared challenges faced by Arctic communities in the context of climate change." And from page 5: "the perception of permafrost-thaw related risks entails significant place- and context-specific complexities." The author(s) allude to some value in comparing results from a pan-Arctic analysis but it needs to be more explicitly stated. The communities included in this study have distinct challenges and situations related to their diverse geo-political contexts and adaptation and planning for risks associated with ongoing and future permafrost thaw will necessarily require a very local approach. I do not suggest that there is no value in comparing and synthesizing data from diverse Arctic regions/communities – just that it would be good to state more clearly from the outset why this is important.

Finally, there are some grammatical errors in this manuscript – the most important for comprehension of the text, I have pointed out below. It is otherwise well written and the figures are relevant and of very high quality.

Page #, line # comments:

Page 2, lines 31-32: with respect to the following sentence: "In addition to rapid environmental change, Arctic regions are characterized by competing geopolitical interests and some undergo rapid societal transformations...", should this be: "... and some are undergoing (or: will undergo or have undergone) rapid societal transformations...?"

Page 2, lines 51-54 and page 5, line 103: Why is it important to include perceptions of risk from diverse geo-political contexts and (Indigenous) local communities...? See comments above under General Comments.

Page 3, section 2.1 plus Figure 2: In this section it is stated that risks arise from the interaction between physical processes impacting or resulting from permafrost thaw; hazards, which are set at the intersection of the natural and societal realms; and societal consequences, corresponding to local perceptions of permafrost thaw impacts, which vary significantly depending on socio-economic, political, environmental, and other contexts. This section also states that Figure 2 depicts risks arising

from permafrost thaw. (The main components of risk are also identified in the Methods, page 15, lines 368-369).

My understanding of the definition of the term, risk, includes the likelihood that a hazard will cause harm and how serious that harm could be. Figure 2 doesn't include any information about how probable it is that the key hazards (middle column) will cause harm nor, at least explicitly, how serious. I see that Figure 3 does show the relevance of the different hazards (from Figure 2) to societal consequences via the use of thicker vs thinner lines, but still feel it would be helpful to more clearly describe the study's own definitions of risk, hazard, consequences, and risk assessment from the outset as these terms are defined differently by researchers in different disciplines. I eventually came to Extended Data Table 1, which does include some of the definitions needed to at least understand Figure 2. As such, Extended Data Table 1 should be referred to earlier on in the manuscript (currently first referred to on page 15, line 363) – perhaps in the figure caption of Figure 2 (something like: 'For the authors' working definitions of physical processes, key hazards, and consequences, see reference #76 and Table 1, Extended Data').

Likewise, the items listed underneath Consequences in figure 2 aren't consequences themselves, but rather broad socio-cultural-health-economic + ecosystem categories delineating important aspects of the arctic communities that I assume will, to varying degrees, experience unwelcome impacts (consequences) from the Key Hazards, which are in turn, related to the Physical Processes. Maybe this is the best, simplest term to use to convey the results of the Thematic Network Analysis in a simple diagram, but a reference to your definition in the text or in the figure caption would help the reader.

Page 6, lines 113-114: I think the word, 'the', is missing from this sentence (i.e. "...practices not adapted to the current climate...").

Page 6, line 119: I think the incorrect figure is referred to here; I do not see a figure of Longyearbyen in the manuscript at all. Also, Figure 5 is referred to in the manuscript before Figure 4.

Page 6, lines 131-132: This sentence is somewhat repetitive: "Rockfalls and landslides are also a major safety concern in Longyearbyen." (compare to lines 128-129, same page).

Page 7, lines 146-147: This sentence is grammatically incorrect and awkward. It is difficult to understand exactly what the author(s) is/are trying to say. Suggest rewriting for clarity.

Page 8, line 161: Here again Figure 5, a photo from Greenland, is referred to when discussing Longyearbyen. Although the theme of ground surface deformation affecting infrastructure fits, it is still a bit confusing when the text is discussing Longyearbyen, but the photo is of Ilulissat (Greenland). It makes more sense to me to refer to Figure 5 two sentences later (lines 162-164), when the deterioration of roads in Avannaata Municipality due to ground surface deformations is discussed.

Page 10, line 214: I believe it is more correct to write 'impacts on health', not, 'impact on health' in this sentence.

General comment (page 13, lines 304-305): This an important point and refreshing to read. In my opinion, high-quality Arctic research that meaningfully integrates traditional knowledge and western science have a pre-requisite of positive, respectful, well-established research relationships between researchers and local communities, individuals, and experts. I am glad to see this written here and hope to see this more in Arctic research literature in the future.

Page 13, lines 310-311: I don't understand what this means: "...including capacity sharing in the context of cross-cultural collaboration and knowledge transfer at the policy and science interface." Suggest adding an example illustrating what this meant in practice to clarify this sentence.

Page 13, lines 321-322: Why weren't natural sciences included here?

Page 13, lines 317-328: I appreciate that the word limit likely doesn't allow for much detail in this regard, but some statement or reference(s) to details on the nature of engagement with the different communities/study sites and the number of engaged participants seems important information for demonstrating how comprehensive and inclusive the representation of perceived permafrost thaw risks across the different study areas was. Some good descriptions of risk workshops from 2023 in Canada, Greenland, and Svalbard are included in the Methods section (pages 16-17), but what about earlier workshops, discussions, interviews, surveys, etc.? Was participation relatively even or quite uneven across the study sites/within individual communities, for example (aside from the Russian sites)? Is there somewhere one could read the survey questions? How many people participated?

Supplementary Information

The Supplementary Info and Extended Data are very helpful, and I would have read them first to improve my initial understanding of the main article had I known. Just a few comments below, first specific, then general.

Page 1, line 20. Is there a number missing here? "...erosion rates of more than m/yr ...".

Page 1, line 24. I am not sure, but should ulus be Ulu?

Page 2, lines 60-65: Perhaps not so important, but I notice some repetitiveness in this section, particularly around permafrost

thawing/active layer increasing.

Page 3, lines 90-92 (and others in this subsection about the Greenland study sites): Some small grammatical and style errors in this section. For example, the word 'plummeting' in the sentence that extends from lines 90-92 is used incorrectly. I suggest just removing it from the sentence, especially since the Longyearbyen temperatures were also described as plummeting in January.

Page 4, line 113: I believe the word 'are', is missing from this sentence (i.e. "...but are changing more slowly...").

Page 4, lines 113-114: Aklavik has very warm ground temperatures compared to all other sites in this study, except Longyearbyen, which have been increasing only slowly. Inuvik is also warm and increasing slowly. Canada shows quite a lot of variability based on the individual communities you have worked with inside the larger Inuvialuit and Gwich'in settlement regions; within these regions as a whole, however, I can see that Russia has clearly experienced a much greater degree of warming. As such, I am not sure I would describe the situation in the way you have here. You could modify your sentence slightly for more accuracy, for example: "...but are changing more slowly compared to Russia and some of the Canadian sites, ...".

General comments

Descriptions of the study regions are noticeably different in content and writing style. There is for example, more historical, political, and economical commentary/description for the Russian communities vs. the Canadian ones. The geomorphological and geological descriptions across the study regions are also uneven (compare Russian and Greenland to Canada, for example). Some of this is addressed later in the document, however, I still recommend editing the text so that it is more systematic and consistent throughout, without being repetitive. The way the effects of climate change across the Canadian study area are listed on page 1 is a good example of how this might be accomplished, which would still allow for more detailed discussion in Supplementary Information 2 without being repetitive.

The description of mean temperature(s) for the different study areas is also uneven. There is no temperature data given for the Russian sites (only for the permafrost) and mean annual temperatures described only for Tuktoyaktuk and Inuvik. Longyearbyen air temperatures, which are warmer than the Canadian and presumably Russian study sites, include mean July temperatures that 'plummet' in January. Longyearbyen also has no real description of its landscape or surficial geology, including glaciers (compare the Greenland description, for example). I appreciate that the same metrics may not be available for all study sites, but this should be noted.

Reviewer #2 (Remarks to the Author):

Dear Authors,

This is a valuable, well-written study that provides an overview of priority risks resulting from Arctic permafrost thaw, based on stakeholder and community workshop data. I find the study to be important, timely, and solid overall, but I do have several comments that may help you refine the final document.

1. After reading the paper, I wonder about deleting the word "comparative" from the title. I see that the study used consolidated data from the four regions, but I missed how it was comparative.
2. In the introduction, I would briefly explain what a risk analysis is / what it is meant to achieve. You could repeat a statement like you have in line 256-257, for example.
3. Check commas throughout paper (e.g. line 165)
4. Please define "thermokarst wetlands or depressions" (line 173) for the unfamiliar reader
5. When a study area is mentioned, I would add the country name in parentheses every time, for the unfamiliar reader.
6. Please define "ice cellars" (line 202) for the unfamiliar reader
7. In lines 283-284 you write "It should be noted that concerns related to permafrost thaw are often conflated with other climate-change related issues on a local level." Can you provide an example of what you mean here and address the implications of this phenomenon?
8. In the conclusion, I suggest adding a call for research to address the "how" of addressing these risks holistically. You outline a host of interrelated challenges, which admittedly left this reader pretty depressed about the enormity of the challenges presented by permafrost thaw without a very clear next step.
9. Line 308: Which perspectives were combined? And how?
10. Line 312: What is meant by "manifold investigations? Is it not possible to quantify and describe the investigations conducted here? I would, at least, include a summary table of the methods used by the different scientific disciplines represented in the study.
11. Line 331: Can you explain how the research approach was "characterized by responsiveness to local research needs and local stakeholder engagement"?
12. Interesting how you averaged local expert rankings with consortium rankings to get final scores. Do you have citations to support this part of the method, or is this a novel aspect of your method?

Best wishes,
Jennifer Holzer

Communications Earth & Environment is committed to improving transparency in authorship. As part of our efforts in this direction, we are now requesting that all authors identified as 'corresponding author' create and link their Open Researcher and Contributor Identifier (ORCID) with their account on the Manuscript Tracking System prior to acceptance. ORCID helps the scientific community achieve unambiguous attribution of all scholarly contributions. You can create and link your ORCID from the home page of the Manuscript Tracking System by clicking on 'Modify my Springer Nature account' and following the instructions in the link below. Please also inform all co-authors that they can add their ORCIDs to their accounts and that they must do so prior to acceptance.

Version 1:

Decision Letter:

Dear Dr Scheer,

Your manuscript titled "Local Risks from Arctic Permafrost Thaw – An Inter- and Transdisciplinary, Comparative Analysis" has now been seen by our reviewers, whose comments appear below. In light of their advice we are delighted to say that we are happy, in principle, to publish a suitably revised version in Communications Earth & Environment.

We therefore invite you to revise your paper one last time to address the remaining concerns of our reviewers. In particular, for publication in Communications Earth & Environment we request that you outline your method and data in details so that other researchers can replicate your work. At the same time, we ask that you edit your manuscript to comply with our format requirements and to maximise the accessibility and therefore the impact of your work.

EDITORIAL REQUESTS:

*****Please take care to match our formatting and policy requirements. We will check revised manuscript and return manuscripts that do not comply. Such requests will lead to delays. *****

SUBMISSION INFORMATION:

OPEN ACCESS:

Communications Earth & Environment is a fully open access journal. Articles are made freely accessible on publication. For further information about article processing charges, open access funding, and advice and support from Nature Research, please visit <https://www.nature.com/commsenv/open-access>

At acceptance, you will be provided with instructions for completing the open access licence agreement on behalf of all authors. This grants us the necessary permissions to publish your paper. Additionally, you will be asked to declare that all

required third party permissions have been obtained, and to provide billing information in order to pay the article-processing charge (APC).

Link Redacted

Best regards,

Martina Grecequet, PhD
Senior Editor,
Communications Earth & Environment
@CommsEarth

REVIEWERS' COMMENTS:

Reviewer #1 (Remarks to the Author):

Dear authors of 'Local risks from Arctic permafrost thaw - an inter- and transdisciplinary, comparative analysis' and editors at Nature Communications, earth & environment:

I have now read through the revised manuscript (#COMMSENV-24-0614A) as well as the Response to Reviewer comments and find that changes to the text (or justification to keep as is), fully address all of my concerns and comments. The new manuscript is easier to follow from beginning to end. The Introduction and Methods sections are in particular, much improved from the original; more transparent and clear for the reader - the inclusion of and rewriting of the descriptions and comparisons between the study areas in particular, is much better. Concrete examples of why a comparative analysis of risks and perceptions of permafrost thaw communities across the circumpolar Arctic in the Results gives the reader a 'red thread' to follow along throughout the text together with the re-written Introduction and Conclusions (e.g. lines 114-116 and 124-126, page 5) and emphasises the importance of this study.

My only additional comments are minor and have to do with language and sentence structure: on page 2 (line 38), I feel it might be better here to have 'as well as' (or just write 'and') before 'ii)' and change 'divergence in' to 'different'. And on page 12, line 240, I wondered if 'a more dominant concern' is better wording than 'preoccupying' (if I understand the meaning of this sentence correctly).

Otherwise, I think this is an important, well-written, well-illustrated, and novel manuscript that should be published in Nature Comm earth & environment as it should appeal to a broad audience. I hope it will inspire more research that fully engages and works together with Arctic communities. I appreciate the opportunity to review it.

Best regards,

Chantel Nixon

Reviewer #2 (Remarks to the Author):

Dear Authors,

For the most part, I found your revision to meet all the reviewers' requests. However, I still feel there is a gap in making your methods replicable. Specifically, you have a section on Preliminary Data Collection (the 42-question quantitative survey), but I didn't find where you explained how you analyzed this and what happened with the results, although this might have been my mistake - I might have missed this.

In the body of the paper, you discuss significant ethnographic research, but I didn't hear how these data were used either.

I found Figure 10 conveying the risk analysis process to be very helpful, but, if possible, I would really like to see an expansion of this figure that includes all data collection stages and all the stages of analysis, with descriptions, so that I can easily trace which types of data went into which types of analysis, and the resulting outputs.

I am sorry to bring this up at this late stage, but I think it's important that this study be replicable and also that the data collected through this study are mentioned, in case others are interested in leveraging them in future research.

Sincerely,

Response to Reviewer 1's Comments

Dear Reviewer,

We sincerely thank you for this thorough review of our manuscript and valuable comments. Please find hereunder our point-by-point response, indicating the changes made to the text and figures.

A red font was specifically used in the PDF version of the revised manuscript to highlight the main edits and corrections. The local risk graphics and study area descriptions, previously included as Extended data and Supplementary materials, have now been integrated within the manuscript.

Sincerely,

The authors.

Reviewer #1 (Remarks to the Author):

Main Article

General Comments

"Local risks from Arctic permafrost thaw – A transdisciplinary, comparative analysis", describes a study of the impacts of permafrost thaw on the environment and the socio-economic, health, and cultural well-being of local rights- and stakeholders in four different Arctic regions (western Canada, Greenland, Svalbard, and Russia). The risk analysis conducted in this study is novel as it considers both physical processes and societal concerns and draws understanding from diverse disciplines and perspectives via multidirectional knowledge exchange processes.

As the authors point out, risks related to permafrost thawing are not perceived or understood uniformly, neither across scientific disciplines nor by local stakeholders, and they therefore prioritized developing a comprehensive and transdisciplinary understanding of both the environmental and societal implications of permafrost thaw. Their analysis reveals five main risks (as well as their causes and consequences) from permafrost thaw, including infrastructure failure, disruptions of mobility and supplies, decrease in water quality, challenges for food security, and exposure to infectious diseases and contaminants.

This study is truly transdisciplinary, combining knowledge and perspectives from a wide variety of experts, both local and non-local, from start to finish. The study regions/communities were specifically chosen according to long-standing, established research relationships, which likely increases participation and adds value.

Local expert and stakeholder knowledge and perspectives have been meaningfully included and presented in the analysis of the risks associated with permafrost thaw – for example, the relevance of the connections between the key hazards and groups of consequences were calculated by averaging the consortium (scientists') and local expert rankings and are clearly indicated in Figure 2 and in Extended Data figures 1-4 by lines of varying thicknesses.

In my specific page #, line # comments below, I recommend that the author(s) provide up-front, clear definitions of important terms in this manuscript, including, risk, hazard, risk assessment, and consequences (while acknowledging the sentence on page 18, line 463 that briefly refers to

the fact that risk definitions (and perceptions) can vary significantly). I also suggest that some additional information about the surveys and workshop participation (within individual communities and comparatively between the study sites), is needed. I looked up reference #76 and see that it addresses both of these particular concerns quite well. Reference #76 (a grant report/deliverable) overlaps with a lot of what is written in the manuscript being reviewed here, but further defines some of the critically important terms such as risk, hazard, etc., and describes the methodology (workshops, surveys, etc.) and results (participation, etc.) in greater detail. My point here is that I am not sure to what degree an article published in a journal such as *Communications Earth & Environment*, with its broad appeal, should require its readers to do so much background reading and/or reading ahead to the Extended Data section to understand some of the key working definitions, figures, and methods presented in the main text. I do feel that clarification of the important terminology, early on in the manuscript, as well as better descriptions of data acquisition methods would help readers better understand the methods and main results of this study and therefore make a bigger impact.

We significantly revised the Introduction, adding the definitions of risk assessment and permafrost-thaw risks adopted in our framework, early on in the manuscript (p2-3).

Assessing risk is now described as the process of systematically identifying, analyzing, and evaluating, either qualitatively or quantitatively, risks (lines 43-44, p2).

In our study, risk is understood as the potential occurrence of a hazard resulting from physical processes and the severity of its consequences for societal domains based on stakeholders' perceptions (lines 57-61, p2). We consider permafrost thaw risks as arising from the interactions between physical processes, hazards and consequences on societal domains. Each of these components is now defined in Introduction from line 61 to 67 (p3), as:

- climatic, environmental and anthropogenic processes contributing to or resulting from permafrost thaw,
- harmful events or dangers, set at the intersection of the natural and societal realms and resulting in adverse consequences for humans, ecosystems and material assets,
- perceived effects or outcomes resulting from a hazard and impacting various societal domains such as health, recreation, the economy and ecosystems.

We additionally elaborated on our understanding of perceptions at lines 60-61 (p2) and 66-67 (p3): *“that is to say the significance assigned to the said risk by stakeholders”* (i.e. the importance of the physical processes in triggering the hazards and of the hazards in impacting societal domains).

These definitions are still provided along with concrete examples in Methods, Table 2 (p26).

We finally acknowledge and discuss the factors and implications of varying risk definitions and perceptions among stakeholders and research disciplines at the following places manuscript.

- Introduction, lines 44-46 and 50-54, page 2: *“In the scientific literature, risk definitions and assessment methods differ greatly [...]”* and *“The fact that risks are not perceived or understood uniformly, neither by local stakeholders nor across scientific disciplines,*

underscores the importance of developing a comprehensive and transdisciplinary understanding [...]”.

- **Conclusion, lines 301-304, page 14:** *“These variations contribute to the complexities in perceived risks related to permafrost thaw. Thus, while the physical processes of permafrost degradation are generally consistent across the study areas, societal consequences and concerns vary significantly due to differing environmental conditions, cultural contexts, and historical legacies.”*
- **Methodological challenges, lines 652-659, page 26:** *“It is important to note that risk definitions and perceptions can vary significantly among scientific disciplines, individuals, and communities. These variations are influenced by a wide array of factors [...]*”.

In order to introduce more information on data acquisition methods and facilitate the comprehension of the study and methodology, information about the conducted survey and workshop participation was added in Introduction (lines 84-89 and 92-93, p3)

Further details regarding the primary data collection and risk ranking workshops are respectively provided in Sections 4.2 (p20) and 4.3.3.2 (p23-24) of the Methods. We notably specified the type of elicitation methods that were adopted (lines 491-495, p20), content of the survey which was conducted in two of the study areas (lines 497-500, p20-21), and number of participants that were involved (lines 493-494, p20; line 500-501, p21; lines 581-582, p23).

I also feel that this manuscript would make a bigger impact if it were to explicitly state the value in comparing results from four very different Arctic regions (i.e. a pan-Arctic analysis). For example, from page 3, lines 66-68: "Such strategies are essential for addressing the challenges posed by permafrost degradation while considering both the unique and shared challenges faced by Arctic communities in the context of climate change." And from page 5: "the perception of permafrost-thaw related risks entails significant place- and context-specific complexities." The author(s) allude to some value in comparing results from a pan-Arctic analysis but it needs to be more explicitly stated. The communities included in this study have distinct challenges and situations related to their diverse geo-political contexts and adaptation and planning for risks associated with ongoing and future permafrost thaw will necessarily require a very local approach. I do not suggest that there is no value in comparing and synthesizing data from diverse Arctic regions/communities – just that it would be good to state more clearly from the outset why this is important.

We agree that the importance of the comparative and synthesizing aspects of our study were not explained and emphasized sufficiently. We added clarification in the Introduction and the Conclusion as indicated hereinafter.

The aims of our study were as follows: 1) assess permafrost thaw risks locally to inform local communities about prominent hazards and consequences within their region (lines 73-75, p3), 2) gain a better understanding of the factors contributing to risks and perceptions by identifying similarities and disparities between study areas characterized by different environmental and societal settings, and thereby facing distinct challenges (line 75, p3), 3) raise general awareness by providing an overview of local permafrost thaw risks and their implications across the Arctic (lines 76-77, p3).

Our comparative approach enabled us to gather a wide range of perceptions and, therefore, to reach a more comprehensive understanding of the interconnected factors of risks in the Arctic (lines 94-97, p3-4; lines 293-297, p14). Identifying similarities and disparities and comparing risks across study areas allowed us to draw conclusions that reach far beyond the particular cases and say something more general about permafrost-thaw related risks in the Arctic. We have not conducted a thorough analytical comparison, analyzing the reasons for similarities and differences between the case studies, but rather a descriptive comparison in order to identify risks related to permafrost thaw for each of the case studies and across them.

As for the composite overview (synthesis), it facilitates a broader understanding of permafrost thaw risks and their implications in the Arctic (lines 76-77, p3).

We finally believe that by comparing and synthesizing risks across various communities and contexts, our findings can be more easily transferred to other (continuous) permafrost regions experiencing similar permafrost-thaw induced risks and sharing similar societal and historical contexts (lines 77-78, p3; lines 304-306, p14-15; lines 327-328, p15). We therefore provide a knowledge basis supporting policy-making and the development of overarching adaptation and mitigation strategies which are needed to manage risks (lines 78-79, p2; line 307, p15; lines 323-324, p15).

Finally, there are some grammatical errors in this manuscript – the most important for comprehension of the text, I have pointed out below. It is otherwise well written and the figures are relevant and of very high quality.

The grammatical errors listed by the Reviewer were corrected as indicated hereunder.

We are additionally planning to seek professional language editing services prior to publication to ensure the quality and correctness of the writing.

Page #, line # comments:

Page 2, lines 31-32: with respect to the following sentence: “In addition to rapid environmental change, Arctic regions are characterized by competing geopolitical interests and some undergo rapid societal transformations...”, should this be: “... and some are undergoing (or: will undergo or have undergone) rapid societal transformations...”?

The sentence was reformulated differently for clarity as *“These risks, in conjunction with rapid socio-environmental transformations and competing geopolitical interests²⁷, necessitate urgent understanding and action”* at lines 26-28 (p1-2).

Page 2, lines 51-54 and page 5, line 103: Why is it important to include perceptions of risk from diverse geo-political contexts and (Indigenous) local communities...? See comments above under General Comments.

We believe it was essential to include diverse risk perceptions, both from a large variety of stakeholders and from diverse geo-political contexts and communities, in order to be able to provide a valid answer to our research question (“What are the local risks from Arctic permafrost thaw?”) and achieve the objectives previously stated.

As explained in our response to “General comments”, we adopted a comprehensive and holistic definition of risks based on perceptions, which are significantly influenced by a wide range of individual and societal factors (lines 36-39, p2; lines 653-659, p26). Assessing risks within a single community, characterized by its unique setting, would not have allowed us to capture a sufficiently broad spectrum of perceptions and their influencing factors. Consequently, by taking into consideration various perceptions and covering diverse contexts and communities, we were able to reach a better understanding of local matters of concerns and identify common patterns among regions (lines 94-97, p3-4; lines 293-297, p14). Hence, we were also able to give a deeper meaning to the rankings attributed by stakeholders to the risks identified in our analysis. Finally, and as previously stated, such a holistic and comparative approach ensures the applicability of the risk assessment framework and transferability of our findings to other regions (lines 77-78, p3; lines 304-306, p14-15; lines 327-328, p15) and supports the development of both locally relevant and overarching coping strategies (lines 78-79, p2; line 307, p15; lines 323-324, p15).

Page 3, section 2.1 plus Figure 2: In this section it is stated that risks arise from the interaction between physical processes impacting or resulting from permafrost thaw; hazards, which are set at the intersection of the natural and societal realms; and societal consequences, corresponding to local perceptions of permafrost thaw impacts, which vary significantly depending on socio-economic, political, environmental, and other contexts. This section also states that Figure 2 depicts risks arising from permafrost thaw. (The main components of risk are also identified in the Methods, page 15, lines 368-369).

My understanding of the definition of the term, risk, includes the likelihood that a hazard will cause harm and how serious that harm could be. Figure 2 doesn't include any information about how probable it is that the key hazards (middle column) will cause harm nor, at least explicitly, how serious. I see that Figure 3 does show the relevance of the different hazards (from Figure 2) to societal consequences via the use of thicker vs thinner lines, but still feel it would be helpful to more clearly describe the study's own definitions of risk, hazard, consequences, and risk assessment from the outset as these terms are defined differently by researchers in different disciplines. I eventually came to Extended Data Table 1, which does include some of the definitions needed to at least understand Figure 2. As such, Extended Data Table 1 should be referred to earlier on in the manuscript (currently first referred to on page 15, line 363) – perhaps in the figure caption of Figure 2 (something like: ‘For the authors’ working definitions of physical processes, key hazards, and consequences, see reference #76 and Table 1, Extended Data’).

As previously explained in our response to “General comments”, our definitions of risk, hazard, consequences, and risk assessment were added in Introduction from line 57 to 67 (p2-3).

The caption of Figure 2 (p5) was supplemented by brief explanations of these terms. Finally, a reference to Table 2 (formerly Extended Data Table 1), which was moved at the end of the Methods section (p26), was also inserted in the caption of Figure 2 as suggested by the Reviewer.

From an engineering or natural science perspective, assessing permafrost thaw risks often entails adopting a probabilistic approach. However, as discussed by Larsen et al., 2021, existing risk assessment frameworks in these scientific disciplines too rarely consider the social and subjective dimensions of risks, i.e., the perceptions of risks. The data we gathered for our risk analysis (cf. Section 4.3.1 in Methods) were purely qualitative, preventing us from using quantitative

assessment techniques. As we aimed to assess risks holistically, we used a semi-qualitative approach to transcribe gathered scientific and non-scientific perceptions into numerical rankings. These rankings are represented in our risk graphics by the thickness of the lines and should be interpreted as explained hereafter. The thicker the connections between the groups of physical processes and key hazards, the most prominent or likely the groups of physical hazards are in triggering the hazard occurrences. Similarly, the thicker the connections between the key hazards and groups of societal domains, the most impacted the societal domains are as a result of the hazards. These explanations were added to the caption of Figure 2 (p5) to facilitate the interpretation of the figure and following local risk graphics (Figures 6 to 9).

Likewise, the items listed underneath Consequences in figure 2 aren't consequences themselves, but rather broad socio-cultural-health-economic + ecosystem categories delineating important aspects of the arctic communities that I assume will, to varying degrees, experience unwelcome impacts (consequences) from the Key Hazards, which are in turn, related to the Physical Processes. Maybe this is the best, simplest term to use to convey the results of the Thematic Network Analysis in a simple diagram, but a reference to your definition in the text or in the figure caption would help the reader.

We agree with this comment. The terms “Consequences on societal domains” or “Impacted societal domains” are now used interchangeably throughout the manuscript and in the risk graphics (Figure 2, p5; Figure 6, p10; Figure 7, p11; Figure 8, p12; Figure 9, p13). We explicitly define what we mean by consequences on societal domains in the Introduction at lines 64-66 (p3) and in the caption of Figure 2 (p5), as suggested by the Reviewer.

Page 6, lines 113-114: I think the word, ‘the’, is missing from this sentence (i.e. “...practices not adapted to the current climate...”).

In order to improve the correctness and meaning of the sentence, it was reformulated as *“Yet, the integrity of the built environment is jeopardized by permafrost thaw and construction practices not adapted to the current climate.”* (lines 135-136, p6).

Page 6, line 119: I think the incorrect figure is referred to here; I do not see a figure of Longyearbyen in the manuscript at all. Also, Figure 5 is referred to in the manuscript before Figure 4.

The order of appearance of the figures and corresponding in-text references, mentioned by the Reviewer, were corrected. Figure 4 (p8) now appears earlier in the manuscript than Figure 5 (p9). In-text references to Figures 4 and 5 were respectively moved to line 157 (p7) and line 185 (p8).

Page 6, lines 131-132: This sentence is somewhat repetitive: “Rockfalls and landslides are also a major safety concern in Longyearbyen.” (compare to lines 128-129, same page).

Redundant information was removed and integrated into the preceding sentence *“Hazardous slope-related processes, including rockfalls and landslides, notably represented a major safety concern in Longyearbyen”* at lines 150-151 (p7).

Page 7, lines 146-147: This sentence is grammatically incorrect and awkward. It is difficult to understand exactly what the author(s) is/are trying to say. Suggest rewriting for clarity.

As suggested by the Reviewer, the sentence was reformulated for clarity as follows: *“The experienced multitude of tasks, difficulties in recruiting and retaining an experienced workforce, resource allocation and prioritizing were perceived as challenging for proactive planning.”* (lines 167-169, p7).

Page 8, line 161: Here again Figure 5, a photo from Greenland, is referred to when discussing Longyearbyen. Although the theme of ground surface deformation affecting infrastructure fits, it is still a bit confusing when the text is discussing Longyearbyen, but the photo is of Ilulissat (Greenland). It makes more sense to me to refer to Figure 5 two sentences later (lines 162-164), when the deterioration of roads in Avannaata Municipality due to ground surface deformations is discussed.

Reference to Figure 5 was moved to line 185 (p8) where road conditions in Greenland are described.

Page 10, line 214: I believe it is more correct to write ‘impacts on health’, not, ‘impact on health’ in this sentence.

We agree with the Reviewer’s suggestion and replaced *“impact on health”* by *“impacts on health”* (line 240, p12).

General comment (page 13, lines 304-305): This an important point and refreshing to read. In my opinion, high-quality Arctic research that meaningfully integrates traditional knowledge and western science have a pre-requisite of positive, respectful, well-established research relationships between researchers and local communities, individuals, and experts. I am glad to see this written here and hope to see this more in Arctic research literature in the future.

Thank you for this positive feedback :)

Page 13, lines 310-311: I don’t understand what this means: *“...including capacity sharing in the context of cross-cultural collaboration and knowledge transfer at the policy and science interface.”* Suggest adding an example illustrating what this meant in practice to clarify this sentence.

The sentence was reformulated as follows for clarity: *“This included capacity sharing, cross-cultural collaboration and knowledge transfer at the policy and science interface¹¹⁹”* (lines 480-481, p20).

Page 13, lines 321-322: Why weren’t natural sciences included here?

Natural sciences were always included, see the sentence before (lines 487-490, p20): *“Within natural and engineering sciences, specific attention was given to permafrost regimes and environmental changes. Geotechnical and geophysical investigations were performed and accompanied by hydrological surveys¹²⁰, soil sampling¹²¹ and remote sensing studies¹²², which were combined with modelling¹²³ and mapping works^{53,124,125}.”* To allow readers to explore the specific methods in more depth, we have included references to some of the most relevant natural and engineering studies conducted within the consortium.

Page 13, lines 317-328: I appreciate that the word limit likely doesn’t allow for much detail in this regard, but some statement or reference(s) to details on the nature of engagement with the different communities/study sites and the number of engaged participants seems important information for

demonstrating how comprehensive and inclusive the representation of perceived permafrost thaw risks across the different study areas was. Some good descriptions of risk workshops from 2023 in Canada, Greenland, and Svalbard are included in the Methods section (pages 16-17), but what about earlier workshops, discussions, interviews, surveys, etc.? Was participation relatively even or quite uneven across the study sites/within individual communities, for example (aside from the Russian sites)? Is there somewhere one could read the survey questions? How many people participated?

In an interdisciplinary project of this scale, quantifying the total volume of data retrieved by the various researchers from different study sites presents significant challenges. Prior to the risk workshops, in Canada, we conducted 51 semi-structured expert interviews and formal discussions, as well as many more informal conversations, with Indigenous land users, permafrost and other local experts. This effort was complemented by the extensive knowledge gathered by a large group of natural scientists from the consortium working in the wider region. In comparison, slightly more interviews, focus groups, and workshops were conducted in the study areas in Greenland (75), Svalbard (80) and Russia (83). Similarly to Canada, informal conversations also provided many additional insights. These methodological details were added to Section 4.2 from line 491 to 495 (p20).

For the risk analysis, we found that the data obtained from Greenland, Canada, and Svalbard were relatively comparable in volume. However, there was a lesser amount of data from Russian sites, as explained at lines 589-591 (p24). This gap was mitigated to some extent through collaboration during the later stages of the risk analysis with researchers who had extensive prior experience working with Russian permafrost communities, thereby enhancing the robustness of our findings despite the initial disparity in data volume.

Supplementary Information

The Supplementary Info and Extended Data are very helpful, and I would have read them first to improve my initial understanding of the main article had I known. Just a few comments below, first specific, then general.

The reviewer should be aware that the content of the Supplementary Information and Extended Data are now included within the core of the manuscript and in the Methods.

Page 1, line 20. Is there a number missing here? "...erosion rates of more than m/yr ...".

The erosion rate was indeed missing from the sentence. The value (10 m/yr) was added at line 363, p16.

Page 1, line 24. I am not sure, but should ulus be Ulus?

The term "ulus" should be written without capital letter. The spelling was retained.

Page 2, lines 60-65: Perhaps not so important, but I notice some repetitiveness in this section, particularly around permafrost thawing/active layer increasing.

Redundant information was removed from the section (lines 403-410, p17-18).

Page 3, lines 90-92 (and others in this subsection about the Greenland study sites): Some small grammatical and style errors in this section. For example, the word ‘plummeting’ in the sentence that extends from lines 90-92 is used incorrectly. I suggest just removing it from the sentence, especially since the Longyearbyen temperatures were also described as plummeting in January.

We agree that the term “plummeting” was wrongly used. The sentences previously containing the term “plummeting” were corrected and reformulated.

To ensure consistency in describing the climatic settings of the study areas, mean annual air temperatures were instead reported for similar reference periods when possible (lines 352-353, p16; lines 382-383, p17; line 400, p17; lines 427-428, p18).

Page 4, line 113: I believe the word ‘are’, is missing from this sentence (i.e. “...but are changing more slowly...”).

The sentence was corrected accordingly at lines 455-456, p19.

Page 4, lines 113-114: Aklavik has very warm ground temperatures compared to all other sites in this study, except Longyearbyen, which have been increasing only slowly. Inuvik is also warm and increasing slowly. Canada shows quite a lot of variability based on the individual communities you have worked with inside the larger Inuvialuit and Gwich’in settlement regions; within these regions as a whole, however, I can see that Russia has clearly experienced a much greater degree of warming. As such, I am not sure I would describe the situation in the way you have here. You could modify your sentence slightly for more accuracy, for example: “...but are changing more slowly compared to Russia and some of the Canadian sites, ...”.

As previously mentioned, the sentence was reformulated to better reflect the differences in warming rates between the study sites (lines 455-456, p19).

General comments

Descriptions of the study regions are noticeably different in content and writing style. There is for example, more historical, political, and economical commentary/description for the Russian communities vs. the Canadian ones. The geomorphological and geological descriptions across the study regions are also uneven (compare Russian and Greenland to Canada, for example). Some of this is addressed later in the document, however, I still recommend editing the text so that it is more systematic and consistent throughout, without being repetitive. The way the effects of climate change across the Canadian study area are listed on page 1 is a good example of how this might be accomplished, which would still allow for more detailed discussion in Supplementary Information 2 without being repetitive.

The descriptions of the study areas (cf. Section 4.1 in Methods, p15) were streamlined and homogenized following the structure of the Longyearbyen case. A similar amount of information regarding the historical/economic/political settings and landscape/geological characteristics was provided for each study area.

The description of mean temperature(s) for the different study areas is also uneven. There is no temperature data given for the Russian sites (only for the permafrost) and mean annual temperatures described only for Tuktoyaktuk and Inuvik. Longyearbyen air temperatures, which are warmer than the Canadian and presumably Russian study sites, include mean July temperatures

that 'plummet' in January. Longyearbyen also has no real description of its landscape or surficial geology, including glaciers (compare the Greenland description, for example). I appreciate that the same metrics may not be available for all study sites, but this should be noted.

As stated previously, the descriptions of the climatic and geological conditions were homogenized across the four study areas (cf. Section 4.1 in Methods, p15). Mean annual air temperatures, ground temperatures and warming rates (both in air and ground temperatures) were provided systematically per study area and for similar reference periods when possible.

Response to Reviewer 2's Comments

Dear Reviewer,

We sincerely thank you for this thorough review of our manuscript and valuable comments. Please find hereunder our point-by-point response, indicating the changes made to the text and figures.

A red font was specifically used in the PDF version of the revised manuscript to highlight the main edits and corrections. The local risk graphics and study area descriptions, previously included as Extended data and Supplementary materials, have now been integrated within the manuscript.

Sincerely,

The authors.

Reviewer #2 (Remarks to the Author):

Dear Authors,

This is a valuable, well-written study that provides an overview of priority risks resulting from Arctic permafrost thaw, based on stakeholder and community workshop data. I find the study to be important, timely, and solid overall, but I do have several comments that may help you refine the final document.

1. After reading the paper, I wonder about deleting the word “comparative” from the title. I see that the study used consolidated data from the four regions, but I missed how it was comparative.

Thank you for this comment, indeed, the comparison aspect warranted clarification. We have now added some additional clarification in the Introduction (lines 68-79, p3) and Conclusion (lines 293-313, p14-15), where we tried to make clear that we have not conducted a thorough analytical comparison, analyzing the reasons for similarities and differences between the case studies. This would have been too lengthy for this paper, and also not necessarily relevant to our research question: “What are the local risks of Arctic Permafrost thaw?” Instead, our comparison was rather descriptive in nature in order to identify risks related to permafrost thaw for each of the case studies and across them.

2. In the introduction, I would briefly explain what a risk analysis is / what it is meant to achieve. You could repeat a statement like you have in line 256-257, for example.

Sentences clarifying the terminology adopted in our study were added to the Introduction at lines 43-44 and 57-67 (p2-3), as suggested by the Reviewers.

Assessing risk is now specifically described as the process of systematically identifying, analyzing, and evaluating, either qualitatively or quantitatively, risks.

3. Check commas throughout paper (e.g. line 165)

The punctuation was proofread throughout the paper and corrected when necessary. We are additionally planning to seek professional language editing services prior to publication to ensure the quality and correctness of the writing.

4. Please define “thermokarst wetlands or depressions” (line 173) for the unfamiliar reader

These terms were defined in parentheses at lines 196-197 (p9).

5. When a study area is mentioned, I would add the country name in parentheses every time, for the unfamiliar reader.

We have now added country names in parentheses once in every section.

6. Please define “ice cellars” (line 202) for the unfamiliar reader

The term “ice cellar” was defined in parentheses at lines 219-220 (p11).

7. In lines 283-284 you write “It should be noted that concerns related to permafrost thaw are often conflated with other climate-change related issues on a local level.” Can you provide an example of what you mean here and address the implications of this phenomenon?

We have reformulated this sentence in the Conclusion (lines 311-313, p15), and tried to clarify further what we meant as follows: *“However, it should be noted that local concerns related to permafrost thaw are deeply entangled with other issues and processes, both those that are climate-related and those that are not, and that any attempt to single out permafrost-thaw related risks necessarily involves an analytical reduction”.*

We also elaborated on this challenge in the Section 4.4 of the Methods (lines 660-665) on page 26: *“Similarly, isolating the impacts of permafrost thaw from other anthropogenic and environmental factors remains difficult. Permafrost risks result from multi-directional and multifactorial relationships, and local observers often did not distinguish between different causes and impacts of changes. These diverse elements, coupled with local complexities in permafrost and geological characteristics, play a pivotal role in shaping how risks are perceived and how communities respond to them”.*

8. In the conclusion, I suggest adding a call for research to address the “how” of addressing these risks holistically. You outline a host of interrelated challenges, which admittedly left this reader pretty depressed about the enormity of the challenges presented by permafrost thaw without a very clear next step.

The last paragraph (lines 314-329, p15) has been significantly revised, adding a sentence (lines 322-324, p15) of how risks can be addressed holistically (essentially by employing holistic, integrated adaptation frameworks), and generally ending on a more positive note, as not to discourage readers in these already challenging times: *“Arctic peoples demonstrate remarkable resilience and adaptability. Adaptation is an ongoing process, as humanity has continuously evolved to meet changing conditions. The inter- and transdisciplinary, composite risk analysis, presented here, provides important insights into the main risks associated with permafrost thaw in Arctic coastal regions and highlights the need for proactive measures to support these*

adaptation and resilience efforts. In some instances, other immediate concerns may overshadow long-term planning. A pragmatic approach allows for focusing on current and emerging prospects and advantages, including, e.g., resource extraction.”

9. Line 308: Which perspectives were combined? And how?

We first briefly mention in the Introduction that both scientific and non-scientific perceptions were specifically integrated in our analysis to assess risks (lines 66-67, p3) and we describe the variety of stakeholders that participated in our study (lines 80-87, p3).

We secondly listed in the Methods (cf. Section 4.2, lines 477-481 and 504-508, p20-21) the different stakeholders that contributed through their perspectives and insights to the primary data collection and gathering for the risk analysis (project consortium and other permafrost scientists, different Indigenous and local knowledge holders).

We used a semi-qualitative approach to transcribe gathered scientific and non-scientific perceptions into numerical rankings. These rankings were attributed by all the stakeholders listed above for the connections between the physical processes, key hazards and impacted societal domains. The combination and averaging of the rankings within and across the study areas is thoroughly detailed in the Methods, Section 4.3.3 (p23) and most particularly Section 4.3.3.3 (p24-25).

9. Line 312: What is meant by “manifold investigations? Is it not possible to quantify and describe the investigations conducted here? I would, at least, include a summary table of the methods used by the different scientific disciplines represented in the study.

In an interdisciplinary project of this scale, quantifying the total volume of data retrieved by the various researchers and describing all the methods adopted in the study areas presents significant challenges. Detailing accurately the different types of site investigations notably carried out for the primary data collection within engineering and natural sciences would be too lengthy. We decided to textually list the most common types of methods and provide references as examples at lines 485-490 (p20). Specific methodological details can be retrieved from the literature published by the project consortium.

Nonetheless, it was possible to further elaborate on the methods used for primary data collection within health, engineering and social sciences (cf. Section 4.2, lines 491-502, p20-21). In Canada, we conducted 51 semi-structured expert interviews and formal discussions - and many more informal conversations - with Indigenous land users, permafrost experts, and other stakeholders. The ethnographic data was collected over a period of six months (fall 2022 to spring 2023), however other members of the consortium with natural science backgrounds have gathered information on risks throughout many years of conducting fieldwork in the wider region. The data for Longyearbyen was collected through 17 months of ethnographic fieldwork over four years (2019-2022). During this time, the researcher actively participated in the town's life, engaged in numerous informal conversations, built relationships, and conducted over 80 audio-recorded sessions, including formal expert, semi-structured, narrative interviews, and focus groups. In the Russian communities, Tiksi and Bykovsky, data were collected through 55 interviews and 3 focus groups. Similarly, 3 workshops and 72 interviews and focus groups were

conducted in Ilulissat and Qaanaaq, Greenland, and were also supplemented by informal conversations. A continuous dialogue with involved stakeholders was notably maintained throughout the study period. The goal was to obtain a diverse and representative sample, including experts on climate change and adaptation (planners, architects, engineers), stakeholders from various sectors (local municipality, tourism industry, Indigenous governments and organizations), and a broader group of inhabitants and visitors.

10. Line 331: Can you explain how the research approach was “characterized by responsiveness to local research needs and local stakeholder engagement”?

We added at lines 511-513 (p21) that it was achieved through intense collaboration with community members: *“This community-based participatory research approach¹⁸ was characterized by its responsiveness to local research needs and stakeholder engagement, achieved by collaborating intensely with community members.”*

We generally mean that the project research questions and methods were designed, iteratively refined and adapted based on local stakeholders’ expressed needs and feedback.

11. Interesting how you averaged local expert rankings with consortium rankings to get final scores. Do you have citations to support this part of the method, or is this a novel aspect of your method?

Best wishes,

Jennifer Holzer

Scientific and non-scientific perceptions are considered as a spectrum rather than entirely distinct viewpoints. The scientists’ perceptions may, for instance, rest on both the findings from their own research and insights provided by local stakeholders. Conversely, the knowledge provided by scientists may, in combination with everyday life experiences and challenges, influence local stakeholders’ perceptions. For this reason, and in order to give an equal weight to both scientific and local perceptions, we averaged the local expert with the consortium ranking (lines 628-629, p25).

No equivalent method averaging scientific and non-scientific perceptions was encountered in the literature within similar research fields. However, a thorough and systematic literature review was not conducted across all research domains.

Response to Reviewer 1's Comments

Dear Reviewer,

We sincerely thank you for taking the time to review our manuscript and for your valuable comments. Please find hereunder our point-by-point response, indicating the changes made to the text.

A red font was used in the newly revised PDF version of the manuscript to highlight edits.

Sincerely,

The authors.

Reviewer #1 (Remarks to the Author):

Main Article

Dear authors of 'Local risks from Arctic permafrost thaw - an inter- and transdisciplinary, comparative analysis' and editors at Nature Communications, earth & environment:

I have now read through the revised manuscript (#COMMSENV-24-0614A) as well as the Response to Reviewer comments and find that changes to the text (or justification to keep as is), fully address all of my concerns and comments. The new manuscript is easier to follow from beginning to end. The Introduction and Methods sections are in particular, much improved from the original; more transparent and clear for the reader - the inclusion of and rewriting of the descriptions and comparisons between the study areas in particular, is much better. Concrete examples of why a comparative analysis of risks and perceptions of permafrost thaw communities across the circumpolar Arctic in the Results gives the reader a 'red thread' to follow along throughout the text together with the re-written Introduction and Conclusions (e.g. lines 114-116 and 124-126, page 5) and emphasises the importance of this study.

Thank you very much.

My only additional comments are minor and have to do with language and sentence structure: on page 2 (line 38), I feel it might be better here to have 'as well as' (or just write 'and') before 'ii)' and change 'divergence in' to 'different'.

We agree with these suggestions and edited the text accordingly at line 81, p3. The manuscript will also undergo language proof-reading offered through a service of the Nature group.

And on page 12, line 240, I wondered if 'a more dominant concern' is better wording than 'preoccupying' (if I understand the meaning of this sentence correctly).

The expression "a more dominant concern" would indeed be better suited in this sentence. We replaced "preoccupying" (line 281, p9).

Otherwise, I think this is an important, well-written, well-illustrated, and novel manuscript that should be published in Nature Comm earth & environment as it should appeal to a broad audience. I hope it will inspire more research that fully engages and works together with Arctic communities. I appreciate the opportunity to review it.

Thanks again for your time and valuable inputs, it is much appreciated!

Best regards,

Chantel Nixon

Response to Reviewer 2's Comments

Dear Reviewer,

We sincerely thank you for taking the time to review our manuscript and for your valuable comments. Please find hereunder our point-by-point response, indicating the changes made to the text.

A red font was used in the newly revised PDF version of the manuscript to highlight edits.

Sincerely,

The authors.

Reviewer #2 (Remarks to the Author):

Dear Authors,

For the most part, I found your revision to meet all the reviewers' requests. However, I still feel there is a gap in making your methods replicable. Specifically, you have a section on Preliminary Data Collection (the 42-question quantitative survey), but I didn't find where you explained how you analyzed this and what happened with the results, although this might have been my mistake - I might have missed this.

In the body of the paper, you discuss significant ethnographic research, but I didn't hear how these data were used either.

The detailed description of the **primary data collection** as described in section 4.3. (natural, engineering and social sciences) is outside the scope of this paper. The methods individually adopted by the different disciplines (e.g. the 42-question survey, interviews conducted as part of the ethnographic fieldwork, etc.) are therefore not described anymore in the Methods. The same is true for the data collected in the engineering, natural and health sciences. We have, however, provided an additional overview of the primary data collection, including the different methods of data collection, types of data collected, and methods of data analysis, along with relevant references (Supplementary Table 1 in Supplementary Information). We have also made a few edits in 4.3. (lines 526-538, p16) and 4.4. to better distinguish between the primary data collection and the risk analysis, the latter being at the heart of this paper.

I found Figure 10 conveying the risk analysis process to be very helpful, but, if possible, I would really like to see an expansion of this figure that includes all data collection stages and all the stages of analysis, with descriptions, so that I can easily trace which types of data went into which types of analysis, and the resulting outputs.

We have provided a more detailed overview over the different stages of the **risk analysis** in the form of detailed flowchart, which includes the requested information (Figure 10, p42) and a detailed table (Supplementary Table 2 in Supplementary Information), including the data collection, analysis, computation and visualization stages.

As previously explained, the primary data collection (section 4.3, p15-16) is outside the scope of this paper. For this reason, the new flowchart mainly put the emphasis on and illustrates the data collection and processing steps that were part of the risk analysis. Additional details are provided in Supplementary Information both for the primary data collection (Supplementary Table 1) and risk analysis (Supplementary Table 2).

I am sorry to bring this up at this late stage, but I think it's important that this study be replicable and also that the data collected through this study are mentioned, in case others are interested in leveraging them in future research.

We greatly appreciate the impetus to more clearly describe our methodology in order to make our research replicable!

Sincerely,

Jennifer Holzer